# Implicit Regularization and Convergence for Weight Normalization

**Xiaoxia Wu**[*]
University of Texas at Austin

**Edgar Dobriban**[*]
University of Pennsylvania

**Tongzhenng Ren**[*]
University of Texas at Austin

**Shanshan Wu**[*]
Google Research

**Zhiyuan Li**
Princeton University

**Suriya Gunasekar**
Microsoft Research

**Rachel Ward**
University of Texas at Austin

**Qiang Liu**
University of Texas at Austin

## Abstract

Normalization methods such as batch [Ioffe and Szegedy, 2015], weight [Salimans and Kingma, 2016], instance [Ulyanov et al., 2016], and layer normalization [Ba et al., 2016] have been widely used in modern machine learning. Here, we study the weight normalization (WN) method [Salimans and Kingma, 2016] and a variant called reparametrized projected gradient descent (rPGD) for overparametrized least squares regression. WN and rPGD reparametrize the weights with a scale $g$ and a unit vector $w$ and thus the objective function becomes *non-convex*. We show that this non-convex formulation has beneficial regularization effects compared to gradient descent on the original objective. These methods adaptively regularize the weights and converge close to the minimum $\ell_2$ norm solution, even for initializations far from zero. For certain stepsizes of $g$ and $w$, we show that they can converge close to the minimum norm solution. This is different from the behavior of gradient descent, which converges to the minimum norm solution only when started at a point in the range space of the feature matrix, and is thus more sensitive to initialization.

## 1 Introduction

Modern machine learning models often have more parameters than data points, allowing a fine-grained adaptation to the data, but also suffering from the risk of over-fitting. To alleviate this, various explicit and implicit regularization methods are used. For instance, weight decay can control the model complexity by shrinking the norm of the weights, and dropout can reduce the model capacity by sub-sampling features during training [Gal and Ghahramani, 2016, Mianjy et al., 2018, Arora et al., 2020]. Recent state-of-the-art techniques such as batch, weight, and layer normalization [Ioffe and Szegedy, 2015, Salimans and Kingma, 2016, Ba et al., 2016], empirically have a regularization effect, e.g., as described in Ioffe and Szegedy [2015], "batch normalisation acts as a regularizer, in some cases eliminating the need for dropout".

While normalization methods are practically popular and successful, their theoretical understanding has only started to emerge recently. For instance, normalization methods make learning more robust to hyperparameters such as the learning rate [Wu et al., 2018, Arora et al., 2019]. Moreover, it has

---

[*]Equal Contribution, xwu@math.utexas.edu, dobriban@wharton.upenn.edu, tongzheng@utexas.edu, shanshanw@google.com

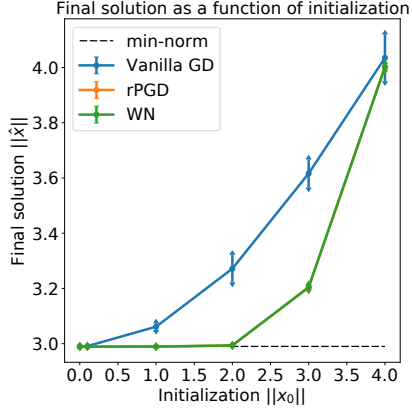

Figure 1: Comparison of the outputs $||\hat{x}|| = ||\hat{g}\hat{w}||$ provided by GD, WN and rPGD on an overparametrized linear regression problem (see Section 1.1). All algorithms (with stepsizes $0.005$) start from the same initialization and stop when the loss reaches $10^{-5}$. Note that the orange (rPGD) and green (WN) curves overlap (see Lemma 2.2 for explanation and Section F for experimental details). GD converges to the minimum $\ell_2$-norm solution only when $||x_0|| = 0$, while WN and rPGD converge close to the minimum norm solution for a wider range of initializations with smaller standard deviation.

been argued that normalization methods can make the model robust to the shift and scaling of the inputs, preventing "internal covariate shift" [Ioffe and Szegedy, 2015] as well as smooth or modify [Santurkar et al., 2018, Lian and Liu, 2019] the optimization landscape.

Yet, a *precise* characterization of the regularization effect of normalization methods in overparametrized models is not available. For overparametrized models, there are typically infinitely many global minima, as shown e.g., in matrix completion [Ge et al., 2016] and neural networks [Ge et al., 2017]. Thus, we can analyze how different algorithms converge to different global minima as a way of quantifying implicit bias. It is critical for the algorithm to converge to a solution with good generalization properties, e.g., Zhang et al. [2016], Neyshabur et al. [2019], etc. For the key model of over-parameterized linear least squares, it is well-known that gradient descent (GD) converges to the minimum Euclidean norm solution when started from zero, [see e.g. Hastie et al., 2019]. It has been argued that this may have favorable generalization properties in learning theory (norms can control the Radamechar complexity), as well as more recent analyses [Bartlett et al., 2019, Hastie et al., 2019, Belkin et al., 2019, Liang and Rakhlin, 2018].

However, for non-convex optimization, starting from the origin might be problematic – this is true in particular in neural networks with ReLU activation function which is often used [LeCun et al., 2015]. In neural networks, we often instead apply random initialization [Glorot and Bengio, 2010, He et al., 2015] which can for instance help escape saddle points [Lee et al., 2016]. Thus, it is important to study algorithms with initializations not close to zero.

With this in mind, we study how a particular normalization method, weight normalization (WN) [Salimans and Kingma, 2016], affects the choice of global minimum in overparametrized least squares regression. WN writes the model parameters $x$ as $x = gw/||w||_2$, and optimizes over the "length" $g \in \mathbb{R}$ and the unnormalized direction $w \in \mathbb{R}^d$ separately. Inspired by weight normalization, we also study a related method where we parametrize the weight as $x = gw$, with $g \in \mathbb{R}$ and a normalized direction $w$ with $||w||_2 = 1$, [see e.g. Douglas et al., 2000]. Different from WN, this method performs projected GD on the unit norm vector $v$, while WN does GD on $w$ such that $w/||w||$ is the unit vector. We call this variant the *reparametrized projected gradient descent* (rPGD) algorithm. We show that the two algorithms (rPGD and WN) have the same limit when the stepsize tends to zero. Arguing in both discrete and continuous time, we show that both find global minima *robust to initialization*.

**Our Contributions.** We consider the overparametrized least squares (LS) optimization problem, which is convex but has infinitely many global minima. As a simplified companion of WN in LS, we introduce the rPGD algorithm (Alg. 2), which is projected gradient descent on a nonconvex reparametrization of LS. We show that WN and rPGD have the same limiting flow—the WN flow—in continuous time (Lemma 2.2). We characterize the stationary points of the loss, showing that the nonconvex reparametrization introduces some additional stationary points that are in general not global minima. However, we also show that the loss still decreases at a geometric rate, if we can control the scale parameter $g$.

How to control the scale parameter? Perhaps surprisingly, we show the delicate property that the scale and the orthogonal complement of the weight vector are linked through an invariant (Lemma 2.5). This allows us to show that the WN flow converges at a geometric rate in spite of the non-convexity

of the reparameterized objective. We precisely characterize the solution, and when it is close to the min norm solution.

In discrete-time, when the stepsize is not infinitely small, we first consider a simple setting where the feature matrix is orthogonal and characterize the behavior of rPGD (Theorem 3.2). We show that by appropriately lowering the learning rate for the scale $g$, rPGD converges to the minimum $\ell_2$ norm solution. We give sharp iteration complexities and upper bounds for the stepsize required for $g$. We extend the result to general data matrices $A$ (Theorem 3.3), where the results become more challenging to prove and a bit harder to parse. This sheds light the empirical observation that only optimizing the direction $w$ training the last layer of neural nets improves generalization [Goyal et al., 2017, Xu et al., 2019].

## 1.1 Setup

We use $\|\cdot\|$ for the $\ell_2$ norm, and consider the standard overparametrized linear regression problem:

$$\min_{x \in \mathbb{R}^d} \frac{1}{2}\|Ax - y\|^2, \tag{1}$$

where $A \in \mathbb{R}^{m \times d}$ ($m < d$) is the feature matrix and $y \in \mathbb{R}^m$ is the target vector. Without loss of generality, we assume that the feature matrix $A$ has full rank $m$. This objective has infinitely many global minimizers, and among them let the minimum $\ell_2$-norm solution be $x^*$. Observe that $x^*$ is characterized by the two properties: (1) $Ax^* = y$; (2) $x^*$ is in the row space of the matrix $A$. We can describe condition (2) via Definition 1.1.

**Definition 1.1.** *For any $z \in \mathbb{R}^d$, we can write $z = z^{\|} + z^{\perp}$ where $Az^{\|} = Az$ and $Az^{\perp} = 0$.*

Then we can equivalently write condition (2) as $x^{*\|} = x^*$. We focus on weight normalization and a related reparametrized projected gradient descent method. Notably, both transform the original convex LS problem to a non-convex problem, which increases the difficulty of theoretical analysis.

**Weight normalization (WN)** WN reparametrizes the variable $x$ as $g \cdot w/\|w\|$, where $g \in \mathbb{R}$ and $w \in \mathbb{R}^d$, which leads to the following minimization problem:

$$\min_{g \in \mathbb{R}, w \in \mathbb{R}^d} h(w, g) = \frac{1}{2}\left\|gAw/\|w\| - y\right\|^2. \tag{2}$$

We can write the min norm solution as $x^* = g^*w^*/\|w^*\|$, where $w^*$ is unique up to scale. However, we can always choose $w^*$ so that $g^* > 0$, unless $x^* = 0$, which implies that $y = 0$. We exclude this degenerate case throughout the paper. The discrete time WN algorithm is shown in Algorithm 1.

| **Algorithm 1** WN for (2) | **Algorithm 2** rPGD for (3) |
|---|---|
| **Input:** Unit norm $w_0$ and scalar $g_0$, iterations $T$, step-sizes $\{\gamma_t\}_{t=0}^{T-1}$ and $\{\eta_t\}_{t=0}^{T-1}$ <br> **for** $t = 0, 1, 2, \cdots, T-1$ **do** <br> $\quad w_{t+1} = w_t - \eta_t \nabla_w h(w_t, g_t)$ <br> $\quad g_{t+1} = g_t - \gamma_t \nabla_g h(w_t, g_t)$ <br><br> **end for** | **Input:** Unit norm $w_0$ and $g_0$, number of iterations $T$, step-sizes $\{\gamma_t\}_{t=0}^{T-1}$ and $\{\eta_t\}_{t=0}^{T-1}$ <br> **for** $t = 0, 1, 2, \cdots, T-1$ **do** <br> $\quad v_t = w_t - \eta_t \nabla_w f(w_t, g_t)$ (gradient step) <br> $\quad w_{t+1} = \frac{v_t}{\|v_t\|}$ (projection) <br> $\quad g_{t+1} = g_t - \gamma_t \nabla_g f(w_t, g_t)$ (gradient step) <br> **end for** |

**Reparametrized Projected Gradient Descent (rPGD)** Inspired by WN algorithm, we investigate an algorithm that directly updates the direction of $w$. See Douglas et al. [2000] for an example of such algorithms. Since the direction is a unit vector, we can perform projected gradient descent on it. To be more concrete, we reparametrize the variable $x$ as $gw$, where $g$ denotes the scale and $w \in \mathbb{R}^d$ with $\|w\| = 1$ denotes the direction, and transform (2) into the following problem:

$$\min_{g \in \mathbb{R}, w \in \mathbb{R}^d} f(w, g) := \frac{1}{2}\|Agw - y\|^2, \quad \text{s.t.} \quad \|w\| = 1. \tag{3}$$

The minimum norm solution can be uniquely written as $x^* = g^*w^*$, where $g^* > 0$ and $\|w^*\| = 1$. To solve (3), we update $g$ with standard gradient descent, and update $w$ via projected gradient descent (PGD) (see Algorithm 2). We call this algorithm reparameterized PGD (rPGD).

One may observe that both algorithms can heuristically be viewed as a variation of adaptive $\ell_2$ regularization, where the magnitude of the regularization depends on the current iteration. We refer the readers to Appendix A for a detailed discussion.

## 2 Continuous Time Analysis

In this section, we study the properties of a continuous limit of WN and rPGD, to give insight into the implicit regularization of normalization methods. We use constant stepsizes for both the update of the scale $g$ and weight $w$, and take them to zero in a way that their ratio remains a constant.

**Condition 2.1** (Stepsizes). *For both Algorithms 1 and 2, use constant stepsizes $\eta_t = \eta$ and $\gamma_t = c\eta$ for $g$ and $w$ respectively, with $c \geq 0$ a fixed constant ratio. We take the continuous limit $\eta \to 0$.*

Setting $c = 0$ amounts to fixing $g$ and only updating $w$. We first prove that the continuous limit of the dynamics of $(g_t, w_t/\|w_t\|)$ for WN evolves the same as the continuous limit of the dynamics of $(g_t, w_t)$ for rPGD, assuming we start with $\|w_0\| = 1$ for WN. The proof can be found in Appendix B.

**Lemma 2.2** (Limiting flow for WN and rPGD). *Assume Condition 2.1 and that $\|w_0\| = 1$ for WN. Then WN (Algorithm 1) with $(g_t, w_t/\|w_t\|)$ and rPGD (Algorithm 2) with $(g_t, w_t)$ have the same limiting dynamics, which we call **WN flow**. This is given by the pair of ordinary differential equations*

$$\frac{dg_t}{dt} = -c\nabla_g f(w_t, g_t) \quad \frac{dw_t}{dt} = -g_t \mathcal{P}_t \left( \nabla_w f(w_t, g_t) \right). \tag{4}$$

*Here $f$ is from (3). With $r = y - Agw$ to denote the residual, $\nabla_w f = A^T r$, $\nabla_g f = w^T A^T r$, and $\mathcal{P}_t = I - w_t w_t^\top / \|w_t\|^2$ the projection matrix onto the space orthogonal to $w_t$.*

While the flow is valuable, the nonconvex reparametrization introduces some new stationary points. We characterize them, and later use this to understand the convergence.

**Lemma 2.3** (Stationary points). *Suppose the smallest eigenvalue of $AA^T$, is positive, $\lambda_{\min} := \lambda_{\min}(AA^T) > 0$. The stationary points of the reparameterized loss from (2) either (a) have loss equal to zero, or (b) belong to the set $\mathcal{S} := \{(g, w) : g = 0, y^T Aw = 0\}$. If the loss (2) at $g, w$ is strictly less than the loss at $(g = 0, w)$, i.e. $\|y\|^2 > \|Agw - y\|^2$, we are always in case (a).*

It is a folklore result that under gradient flow, the loss is non-increasing even in the nonconvex case [see e.g. Rockafellar and Wets, 2009]. For the WN gradient flow, we can make this folklore rigorous and, provided the scale parameter $g_t$ is lower bounded, show that the loss decreases at a *geometric rate*.

**Lemma 2.4** (Rate of $\|r_t\|$). *Under the setting of Lemma 2.2, we have the bounds:*

$$-\max\{g_t^2, c\}\|A^T r_t\|^2 \leq d[1/2\|r_t\|^2]/dt \leq -\min\{g_t^2, c\}\|A^T r_t\|^2 \leq 0. \tag{5}$$

*This shows that $\|r_t\|$ is non-increasing. If for some $C > 0$, $g_t > C$ for all $t$, then the loss decreases geometrically at rate $\min(C^2, c)$.*

How can we control the scale parameter? Perhaps surprisingly, we show that the scale parameter and the orthogonal complement of the weight vector are linked through an *invariant*.

**Lemma 2.5** (Invariant). *Assume $c > 0$ in Condition 2.1. Under the setting from Lemma 2.2, let $w_t = w_t^\perp + w_t^\|$ as defined in Definition 1.1. We have at time $t > 0$,*

$$w_t^\perp = \exp\left( \frac{g_0^2 - g_t^2}{2c} \right) w_0^\perp \quad \text{and so} \quad \|w_t^\perp\|^2 \cdot \exp(g_t^2/2c) = \|w_0^\perp\|^2 \cdot \exp(g_0^2/2c). \tag{6}$$

Lemma 2.5 shows that the orthogonal complement $w_t^\perp$ can change during the WN flow dynamics. This is the key property of WN that can yield additional regularization. Lemma 2.5 also implies that $\|w_t^\perp\|^2 \cdot \exp(g_t^2/2c)$ is invariant along the path. If we initialize with small $|g_0|$ and $|g_t|$ is greater than $|g_0|$ (we will describe the dynamics of $g_t$ in the next part), then $\|w_t^\perp\|^2$ will decrease, and we get close to the minimum norm solution. This is in contrast to gradient descent and flow, where $\|w^\perp\|^2$ is preserved (see e.g., [Hastie et al., 2019]).

The invariant (6) in the optimization path holds for certain more general settings. Specifically, it holds for linearly parametrized loss functions that only depend on a small dimensional linear subspace of the parameter space (e.g., overparametrized logistic regression). See Appendix D. Equipped with the above lemmas, we can discuss the solution and implicit regularization effect of the WN flow.

**Theorem 2.6** (WN flow Solution). *Assume Condition 2.1 and $\lambda_{\min} > 0$. Suppose we initialize the WN flow at $g_0, w_0$, such that $\|w_0\| = 1$. We have that either (a) the loss converges to zero, or (b) the iterates $(g_t, w_t)$ converge to a stationary point in $\mathcal{S}$ as defined in Lemma 2.3. In case (a), we characterize the solutions based on $g_t$:*

> **Part I.** *If $c > 0$, and the loss converges to zero, the solution can be expressed as*
>
> $$\lim_{t \to \infty} g_t w_t = x^* + g^* w_0^\perp \exp\left(\frac{g_0^2 - g^{*2}}{2c}\right). \tag{7}$$
>
> *A sufficient condition for the loss converging to zero is that $\|y\|^2 > \|Ag_0 w_0 - y\|^2$.*

> **Part II.** *If $c = 0$ and $A$ is orthogonal, i.e., $AA^T = I$, then $w_t \to w^*$. If $A$ is not orthogonal, then the flow still converges to a point $\tilde{w}_0$ in the row space of $A$ (i.e, $\tilde{w}_0^\perp = 0$). When restarting the WN flow with $c > 0$ from $g_0, \tilde{w}_0$, then $(g_0, \tilde{w}_0) \to (g^*, w^*)$.*

We defer the proofs of Lemmas 2.4, 2.5 and Theorem 2.6 to Appendix C. Part I of Theorem 2.6 shows that, if we initialize with $g_0^2 \le g^{*2}$ and we are not stuck at $\mathcal{S}$, the WN flow will converge to a solution that is close to the minimum norm solution. Compared with GD where the final solution is $x_t = x^* + g^* w_0^\perp$, WN flow has smaller component in the orthocomplement of the row space of $A$. In contrast, if $g_0^2 > g^{*2}$, then WN flow can converge to a solution that is *farther* from $x^*$ than GD.

Part II in Theorem 2.6 shows a distinction between orthogonal and general $A$. For orthogonal $A$, even fixing the scale $g_0$ we can converge to the direction of the minimum norm solution. Although we do not directly recover $g^*$ in the flow, this can be recovered as $|g^*| = \|y\|$. For general $A$ with fixed $g$, we do not necessarily converge to the right direction, only to the row span of $A$. However, if we run the flow with $c = 0$ until convergence, and then turn on the flow for $g$ (i.e. set $c > 0$), we converge to the minimum norm solution. The results for discrete time presented later mirror this. See Figure 2 for an illustration. We mention that the flow for the fixed $g$ case is well known [See e.g. Helmke and Moore, 2012, Section 1.6]), in the special case that the matrix $A$ is square.

Theorem 2.6 provides no rate of convergence. By our results on the rate of $\|r_t\|$, and by controlling $g_t$ using the invariant, we can provide a convergence rate below. The following theorem has two convergence rates, depending on the magnitude of $g_0$ and on whether we initialize $g_0$ and $w_0$ with the initial loss smaller than the loss at zero or not. If both rates are valid for a certain parameter configuration, then the faster of the two applies.

**Theorem 2.7** (Convergence Rate). *Assume Condition 2.1, $c > 0$ with $\|w_0\| = 1$ and the smallest eigenvalue $\lambda_{\min}$ of $AA^T$ is strictly positive.*

- *If $g_0^2 > 2c \log(1/\|w_0^\perp\|)$, the loss along the WN flow path $(g_t, w_t)$ decreases geometrically, satisfying $f(w_T, g_T) \le \varepsilon$ after time*

$$T = \frac{\log(f(w_0, g_0)/\varepsilon)}{\lambda_{\min} \min\left\{2c \log \|w_0^\perp\| + g_0^2, c\right\}}.$$

- *If the initial loss smaller than the loss at zero, $\delta = (\|y\|^2 - \|Ag_0 w_0 - y\|^2)/\lambda_{\max} > 0$, then $f(w_T, g_T) \le \varepsilon$, after time*

$$T = \frac{\log(f(w_0, g_0)/\varepsilon)}{\lambda_{\min} \min\{\delta, c\}} + \frac{1}{\lambda_{\max}} \log\left(2 - \frac{g_0}{\delta}\right) \mathbb{1}_{\{g_0 < \delta\}}.$$

The two convergence rates apply to somewhat complementary cases. In the first case, it follows from the invariant that as long as $g_0$ is above the required threshold $2c \log(1/\|w_0^\perp\|)$, the loss converges geometrically. Otherwise, if we have $\delta > 0$ (that is, we initialize below the loss at zero), the dynamics of $g_t^2$ turn out to have a favorable "self-balancing" geometric property, i.e., they start to increase when they get sufficiently small (c.f. equation (14) in Lemma C.1), and we can also get the geometric convergence, instead of being stuck at $\mathcal{S}$. The theorem only shows convergence, not implicit regularization. As described above, the regularization is favorable if $|g_0| < |g^*|$.

**A Concrete Example.** To gain more insight, we provide here a simple example (see also Figure 2). Suppose we have a two-dimensional parameter $w$, and we make a 1-dimensional observation using the matrix $A = [1, 0]$, and $y = 1$. Then, the equation we are solving is $gw[1] = 1$ (where square

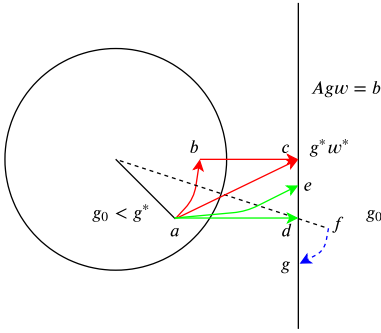

Figure 2: Consider the function $f(w_1, w_2, g)$ with $A = [1, 0] \in \mathbb{R}^{1 \times 2}$. Suppose we start with $g_0 < g^*$ (point $a$). Then GD converges to $d$, while rPGD and WN could result in a point ($e$ or $c$) closer to minimum norm $c$ depending on the stepsize schedule of $g$. Part I in Theorem 2.6 suggests that rPGD and WN will follow the path $a \to e$ if $\gamma_t$ and $\eta_t$ converge to zero at the same rates, and Part II implies the red path $a \to b \to c$ to the minimum norm solution (if $g_0$ is fixed for a certain time, and updated later). The optimal path $a \to c$ is taken when $g$ is updated in a careful way. On the other hand, starting with $g_0 > g^*$, for instance at $f$, (7) shows the limit is $g$, further away from $g^* w^*$.

[1]The figure is for $f(w_1, w_2, g) = (gw_1 - 2)^2$, with minimum norm solution at $c = (2, 0)$.

brackets index coordinates of vectors), and the minimum norm solution is $w = [1, 0]^T$, with $g^* = 1$. Our results guarantee that the WN flow converges to either (1) a zero of the loss, or (2) to a stationary point such that $g = 0$ and $w^T A^T y = 0$. The second condition reduces to $w[1] \cdot y = 0$. Now, if $y \neq 0$ (which is the typical case), then this reduces to $w[1] = 0$, and since $\|w\| = 1$, we have two solutions $w[2] = \pm 1$. So this leads to two spurious stationary points $(g, w) = (0, [0, \pm 1]^T)$, which are not global minima. The loss value at these points is $1$, and so if we start at any point such that the loss is less than one, then we converge to a global minimum. If $y = 0$, then this leads to infinitely many stationary points, i.e. all of those with $g = 0$, but these turn out to be global minima.

Suppose moreover that we start with $w_0 = [0, 1]^T$ and set $c = 1$. Suppose now that we start with some $g_0 \neq 0$. Then WN flow converges to a solution $gw = [1, \exp([g_0^2 - 1]/2)]^\top$. If $g_0$ is relatively small, this quantity is close to $x^* = [1, 0]^\top$, closer than the gradient flow solution $[1, 1]$.

## 3 Discrete Time Analysis

In this section, we switch to discrete time. It turns out that analyzing rPGD is more tractable than WN, so we will focus on rPGD. Since the two algorithms collapse to the same flow in continuous time, their dynamics should be "close" in finite time, especially in the small stepsize regime. We show that rPGD with properly chosen learning rates converges close to the min norm solution even when the initialization is *far away from the origin*. We study rPGD based on the intuition that $\|w_t^\perp\|$ decreases after the normalization step.

**Orthogonal Data Matrix.** Consider first the simple case where the feature matrix $A$ has orthonormal rows, i.e., $AA^\top = I$. Our strategy for rPGD to reach the minimum $\ell_2$-norm solution is *to use the optimal stepsize for $w$ and a small stepsize for $g$ such that $g_t^2 < g^{*2}$ for all iterations*. The key intuition is that with a small stepsize, the loss stays positive and ensures the direction $w_t$ has sufficient time to find $w^*$. On the contrary, if we use a large stepsize for $g$, then it is possible for $g_t$ to be greater than $g^*$ so that $w_t$ can potentially converge to the wrong direction.

**Condition 3.1.** *(Two-stage learning rates) We update $w$ with its optimal step-size $\eta_t = 1/g_t^2$.[2] For the stepsize of $g_t$, we use two constant values: (a) $\gamma_t = \gamma^{(1)}$ when $0 \leq t \leq T_1$; (b) $\gamma_t = \gamma^{(2)}$ when $t \geq T_1 + 1$, for a $T_1$ specified below.*

**Theorem 3.2** (Convergence for Orthogonal Matrix $A$). *Suppose the initialization satisfies $0 < g_0 < g^*$, and that $w_0$ is a vector with $\|w_0\| = 1$. Let $\delta_0 = (g^*)^2 - (g_0)^2$. Set an error parameter $\varepsilon > 0$ and the stepsize given in Condition 3.1 with a hyper-parameter $\rho \in (0, 1]$ for $\gamma^{(1)}$. Running the rPGD algorithm, we can reach $\|w_{T_1}^\perp\| \leq \varepsilon$ and $g_{T_1}^2 \leq g^{*2} - \rho \delta_0$ after $T_1$ iterations, and $\|w_T^\perp\| \leq \varepsilon$ and $\|Ag_T w_T - b\|^2 \leq 3\varepsilon g^{*2}$ after $T = T_1 + T_2$ iterations, if we set stepsizes as follows.*

[2]The Hessian for $w$ in problem 3 is $\nabla_w^2 f(w, g) = g^2 A^\top A$. For orthogonal $A$, $\lambda_{\max}(\nabla_w^2 f(w, g)) = g^2$.

*(a)* Set $\gamma^{(1)} = \mathcal{O}\left(\frac{\rho}{\log(1/\varepsilon)} \left(\frac{g_0}{g^*}\right)^2 \log\left((1-\rho)\frac{g^*}{g_0} + \rho\right)\right)$ and $\gamma^{(2)} \le \frac{1}{4}$. Then we have

$$T_1 = \mathcal{O}\left(\frac{(g^*)^2}{\rho\delta_0} \log\left(\frac{1}{\varepsilon}\right)\right); \quad T_2 = \mathcal{O}\left(\frac{1}{\gamma^{(2)}} \log\left(\frac{(\rho\delta_0/g^{*2})^2}{\varepsilon}\right)\right).$$

*(b)* Set $\gamma^{(1)} = 0$ and $\gamma^{(2)} < \frac{1}{4}$. Then we have

$$T_1 = \mathcal{O}\left(\frac{g_0^2}{\delta_0} \log\left(\frac{1}{\varepsilon}\right)\right); \quad T_2 = \mathcal{O}\left(\frac{1}{\gamma^{(2)}} \log\left(\frac{\sqrt{\delta_0/g^{*2}}}{\varepsilon}\right)\right).$$

We restate the theorem with the explicit forms of $T_1$ and $T_2$, along with the proof, in Appendix E.1 and E.2. The theorem requires knowing $g^*$, which can be approximated by $\|y\|$ (as $g^* = \|y\|$). When all parameters other than $\varepsilon$ are treated as constants, this shows that rPGD converges to the minimum norm solution with the same rate $\log(1/\varepsilon)$ as standard GD starting from the origin. However, the constants in front the $\log(1/\varepsilon)$ can be large: e.g., in case (a), $(g^*)^2/\rho\delta_0$ can be $\gg 1$ if $\rho$ is small or if $|g_0|/|g^*| \approx 1$. This first $T_1$ iterations allow $w_t$ to "find" $w^*$, while the remaining $T_2$ allow $g_t$ to converge to $g^*$. Both cases show an intrinsic tradeoff between $T_1$ and $T_2$: a larger $\delta_0$ (being far from $g^*$) leads to faster convergence in the first phase for $\|w_t^\perp\|$ (i.e. smaller $T_1$), but slower convergence in $g_t$ and loss (i.e. larger $T_2$). Specifically, notice that $\delta_0$ is in the denominator of $T_1$ but in the numerator of $T_2$.

Our proof shows that $g_t$ is always increasing for any $g_0 > 0$ (c.f. Lemma E.4). Moreover, $\|w_t^\perp\|$ decreases at a geometric rate, $\|w_{t+1}^\perp\|^2 \le (g_t^2/g^{*2})\|w_t^\perp\|^2$, as long as $|g_t|$ is not too close to $|g^*|$ (c.f. Lemma E.6 or Equation (19)). This is why the condition $g_{T_1}^2 \le g^{*2} - \rho\delta_0$ is needed, ensuring that $g_t$ is far away from $g^*$ in all steps before $T_1$. This is also why we require a stepsize for $g_t$ of order $1/\log(1/\varepsilon)$ (c.f. Equation 20), which is smaller than the constant stepsize in usual GD. Here $\rho$ leads to a tradeoff between $T_1$ and $T_2$: a smaller $\rho$ results in larger $T_1$ but smaller $T_2$, vice versa. When $\rho \approx \varepsilon$, we have $T_1 = \mathcal{O}(1/\varepsilon)$ up to log factors, (a slower rate) and $T_2 = \mathcal{O}(1)$. [3] A constant order $\rho$ leads to a faster $\log(1/\varepsilon)$ rate. However, we choose to state the result for the entire range of $\rho \in (0, 1]$ for completeness. When $\rho = 1$, the stepsize $\gamma^{(1)}$ becomes zero, hence $g_t$ does not change. In this case, we can get a stronger result for $T_1$ (stated in (b)) using a slightly different method of proof, improving the bounds of case (a) respectively with a factor of $(g^*/g_0)^2 > 1$ for $T_1$.

We remark that for orthogonal $A$ with the optimal stepsize $(1/g_t^2)$ for $w_t$ and $g_0 \ne 0$, we have $Aw_{t+1} = Ag^*w^*/(g_t\|v_t\|) \ne 0$ (c.f. Lemma E.2). Thus we can escape the saddle points and reach the global minimum, unlike in continuous time where we can be stuck at the stationary points $\mathcal{S}$.

We reiterate that our motivation is not to outperform other methods (e.g. GD starts from zero) in search of the minimum norm solution, but to characterize the regularization effect of weight normalization and shed light on the empirical observation that fixing the scalar $g$ and only optimizing the directions $w$ in training the last layer of neural networks can improve generalization [Goyal et al., 2017, Xu et al., 2019]. This is, to our knowledge, the first kind of theory on how to control the learning rates of parameters in weight normalization such as to converge to minimum norm solutions *for initialization not close to origin*, which may have beneficial generalization properties.

**General Data Matrix.** Inspired by the analysis for orthogonal $A$, we now study general data matrices. As we have seen from the orthogonal $A$ case, the stepsize for the scale parameter should be extremely small or even 0 to make $\|w_t^\perp\|$ small. Thus, for simplicity, we focus on fixing $g := g_0$ and update only $w$ using rPGD so that the orthogonal component $w^\perp$ decreases geometrically until $\|w_{T_1}^\perp\| \le \varepsilon$. In addition, we notice from the analysis in Theorem 3.2 that updating $g_t$ and $w_t$ separately after $t > T_1$ (i.e., reaching small $\|w_t^\perp\|$) shows no advantage over GD using $x = gw$. Thus, the best strategy to find $g^*w^*$ is to use rPGD only updating $w_t$ (so $g_t = g_0 < g^*$) and then apply standard GD after $T_1$ once we have $\|w_{T_1}^\perp\| \le \varepsilon$. We focus on the complexity of $T_1$ in the remainder, as the remaining steps are standard GD, which is well understood.

Even though we fix $g$, the problem is still non-convex because the projection is on the *sphere* (rather than the *ball*), a non-convex surface. However, suppose we can ensure that after each update, the

gradient step $v_t = w_t - \eta_t \nabla_w f(w_t, g_t)$ has norm $\|v_t\| \geq 1$. Then the following two constrained non-convex problems are equivalent:

$$\min_{w \in \mathbb{R}^d} \|Ag_0 w - y\|^2 \text{ s.t. } w \in \{w, \|w\| = 1\} \quad \Leftrightarrow \quad \min_{w \in \mathbb{R}^d} \|Ag_0 w - y\|^2 \text{ s.t. } w \in \{w, \|w\| \leq 1\}$$

Thus our analysis will focus on showing that $\|v_t\| \geq 1$. Note that, without loss of generality we can always scale $A$ so that its largest singular value is one.

**Proposition 3.3** (General Matrix $A$). *Fix $\delta > 0$, and fix a full rank matrix $A$ with $\lambda_{\max}(AA^\top) = 1$. With a fixed $g = g_0$ satisfying $g_0 \leq [g^* \lambda_{\min}(AA^\top)]/(2+\delta)$, we can reach a solution with $\|w_{T_1}^\perp\| \leq \varepsilon$ in a number of iterations*

$$T_1 = \log\left(\frac{\|w_0^\perp\|}{\varepsilon}\right) / \log(1 + \delta).$$

The proof is in Appendix E.3. The proposition implies that if we set a small $g_0 = \mathcal{O}(g^* \lambda_{\min}(AA^\top))$ for general $A$ and $w^*$, running rPGD with fixed $g_0$ helps regularize the iterates. After starting from $w_{T_1}$, we can converge close to the minimum norm solution using standard GD. If the eigenvalues of $A$ are "not too spread out", we can get a better condition for $g_0$ using concentration inequalities for eigenvalues. See inequality (50) in Proposition E.9 for more details.

## 4 Discussion

**Limitation of our work.** It is important to recognize the limitations of our work. First, our theoretical work only addresses weight normalization (not batch, layer, instance or other normalization methods), and only concerns the setting of linear least squares regression. While this may seem limiting, it is still significant: even in this setting, the problem is not understood, and leads to intriguing insights. In fact there is some recent work on Neural Tangent Kernels arguing overparametrized NNs can be equivalent to linear problems, see e.g., [Jacot et al., 2018, Du et al., 2018, Lee et al., 2019], etc. Second, the continuous limit is only an approximation; however it leads to elegant and interpretable results, which are moreover also reflected in simulations. Third, some of our results concern a two-stage algorithm where the scale is fixed for the first stage; nevertheless, our results on the standard "one-stage" algorithm in continuous time suggest such discrete-time results extend to the situation where the scale is not fixed, but slowly-varying for the first stage.

**Related Work.** While there is a large literature on weight normalization and implicit regularization (see Sec 4), our work differs in crucial ways. We study the overparametrized case and characterize the implicit regularization for a broad range of initializations (unlike works that study initialization with small norm). Also, we prove convergence and characterize the solution explicitly (unlike works such as [Gunasekar et al., 2018] that assume convergence to minimizers). Below we can only discuss a small number of related works.

*Implicit regularization.* It has been recognized early that optimization algorithms can have an implicit regularization effect, both in applied mathematics [Strand, 1974], and in deep learning [Morgan and Bourlard, 1990, Neyshabur et al., 2014]. It has been argued that "algorithmic regularization" can be one of the main differences between the perspectives of statistical data analysis and more traditional computer science [Mahoney, 2012].

Theoretical work has shown that gradient descent is a form of regularization for exponential-type losses such as logistic regression, converging to the max-margin SVM for separable data [Soudry et al., 2018, Poggio et al., 2019], as well as for non-separable data [Ji and Telgarsky, 2019]. Similar results have been obtained for other optimization methods [Gunasekar et al., 2018], as well as for matrix factorization [Gunasekar et al., 2017, Arora et al., 2019], sparse regression [Vaškevičius et al., 2019], and connecting to ridge regression [Ali et al., 2018]. For instance, [Li et al., 2018] showed that GD with small initialization and small step size finds low-rank solutions for matrix sensing. There have also been arguments that neural networks perform a type of self-regularization, some connecting to random matrix theory [Martin and Mahoney, 2018, Mahoney and Martin, 2019]. Popular methods for regularization include weight decay (a.k.a., ridge regression) [Dobriban and Wager, 2018, Liu and Dobriban, 2019], dropout [Wager et al., 2013], data augmentation [Chen et al., 2019], etc.

*Convergence of normalization methods.* [Salimans and Kingma, 2016] argued that their proposed weight normalization (WN) method, optimizing $x = gw/\|w\|_2$ over $g \geq 0$ and $w \in \mathbb{R}^d$, increases

the norm of $w$, and leads to robustness to the choice of stepsize. [Hoffer et al., 2018] studied normalization with weight decay and learning-rate adjustments. [Du et al., 2018] proved that GD with WN from randomly initialized weights could recover the right parameters with constant probability in a one-hidden neural network with Gaussian input. [Ward et al., 2019] connected the WN with adaptive gradient methods and proved the sub-linear convergence for both GD and SGD. [Cai et al., 2019] showed that for under-parametrized least squares regression (which is different from our over-parametrized setting), batch normalized GD converges for arbitrary learning rates for the weights, with linear convergence for constant learning rate. Similar results for scale-invariant parameters can be found in [Arora et al., 2018] with more general models, extending to the non-convex case. [Kohler et al., 2019] proved linear convergence of batch normalization in halfspace learning and neural networks with Gaussian data, using however parameter-dependent learning rates and optimal update of the length $g$. [Luo et al., 2019] analyzed batch normalization by using a basic block of neural networks and concluded that batch normalization has implicit regularization. [Dukler et al., 2020] discussed the convergence of two-layer ReLU network with weight normalization under the NTK regime. However, none of the above give the invariants we do.

*Nonlinear Least Mean Squares (NLMS)*. Normalization methods are possibly related to the Nonlinear Least Mean Squares (NLMS) methods from signal processing [see e.g. Proakis, 2001, Haykin and Widrow, 2002, Haykin, 2005, Hayes, 2009]. NLMS can be viewed as an online algorithm where the samples $a_t, y_t$ ($a_t$ are the rows of $A$, $y_t$ are the entries of $y$) arrive in an online fashion, and we update the iterates as $x_{t+1} = x_t - \eta r_t a_t / \|a_t\|^2$, where $r_t = y_t - x_t^\top a_t$ are the residuals. There is a connection to randomized Kaczmarcz methods Strohmer and Vershynin [2009]. However, it is not obvious how they are related to weight normalization or rPGD/WN, e.g., these methods are under online setting, while rPGD/WN are offline.

## Broader Impact

Our work is on the foundations and theory of machine learning. One of the distinctive characteristics of contemporary machine learning is that it relies on a large number of "ad hoc" techniques, that have been developed and validated through computational experiments. For instance, the optimization of neural networks is in general a highly nonconvex problem, and there is no complete theoretical understanding yet as to how exactly it works in practice. Moreover, there a large number of practical "hacks" that people have developed that help in practice, but lack a solid foundation. Our work is about one of these techniques, weight normalization. We develop some nontrivial theoretical results about it in a simplified "model". This work does not directly propose any new algorithms. But we hope that our work will have an impact in practice, namely that it will help practitioners understand what the WN method is doing (important, as people naturally want to understand and know "why" things work), and possibly in the future, help us develop better algorithms (here the principle being that "if you understand it you can improve it", which has been useful in engineering and computer science for decades).

## Acknowledgments

The authors thank Nathan Srebro and Sanjeev Arora for constructive suggestions. XW, ED, SG, and RW thank the Institute for Advanced Study for their hospitality during the Special Year on Optimization, Statistics, and Theoretical Machine Learning. XW, SW, ED, SG, and RW thank the Simons Institute for their hospitality during the Summer 2019 program on the Foundations of Deep Learning. RW acknowledges funding from AFOSR and Facebook AI Research. This material is based upon work supported by the National Science Foundation under Grant No. DMS-1638352.

## Footnotes

[3]Note that the bound for $T_1$ could be tightened, possibly to $\log(1/\varepsilon)$, by using refined analysis at the step from (18) to (19).

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
