[Supplementary Material]

# Content

# A  Adaptive Regularization

We illustrate that the two Algorithms can heuristically be viewed as GD on adaptively $\ell_2$-regularized regression problems. The regularization parameter changes for each iteration in the algorithms:

$$\min_{x_{t+1}} \left( \frac{1}{2} \|Ax_{t+1} - Ax^*\|^2 + \lambda(w_t, g_t, \eta, A, y)\|x_{t+1}\|^2 \right).$$

However, it is difficult to characterize the behavior of $\lambda_t$ in general.

**WN.** Let $x_t = \frac{g_t w_t}{\|w_t\|}$. Notice that:

$$\frac{g_{t+1} w_{t+1}}{\|w_{t+1}\|} = \frac{g_t}{\|w_t\|} w_t + \left( \frac{g_{t+1}}{\|w_{t+1}\|} - \frac{g_t}{\|w_t\|} \right) w_t - \eta_t \frac{g_{t+1}}{\|w_{t+1}\|} \frac{g_t}{\|w_t\|} (I - \frac{w_t w_t^T}{\|w_t\|^2}) A^T r$$

This can be translated to the update of $x_t$ as

$$x_{t+1} = x_t - \eta_t \frac{g_{t+1} g_t}{\|w_{t+1}\|\|w_t\|} A^T r - \left( 1 - \frac{g_{t+1}\|w_t\|}{g_t\|w_{t+1}\|} - \eta_t \frac{g_{t+1}}{\|w_{t+1}\|} \langle \frac{w_t}{\|w_t\|^2}, A^T r \rangle \right) x_t$$

**rPGD.** Let $x_t = g_t w_t$. The update of $w_t$ in Algorithm 2 is

$$w_{t+1} = \frac{1}{\|v_t\|} (w_t - \eta_t g_t A^T (Ax_t - Ax^*)). \tag{8}$$

We can now write the update of $x_{t+1} = g_{t+1} w_{t+1}$ as

$$x_{t+1} = x_t - \frac{\eta_t g_t g_{t+1}}{\|v_t\|} A^T (Ax_t - Ax^*) - (1 - \frac{g_{t+1}}{g_t\|v_t\|})x_t.$$

Both updates can be viewed as a gradient step on the following $\ell_2$-regularized regression problem, with specific choices of $\lambda_t$ at iteration $t$:

$$\frac{1}{2} \|Ax_{t+1} - Ax^*\|^2 + \lambda_t \|x_{t+1}\|^2,$$

We see that the regularization parameter changes for each iteration for both WN and rPGD, as follows:

$$(\text{WN}) \quad \lambda_t = \frac{\|w_{t+1}\|\|w_t\|}{g_{t+1} g_t} (1 - \frac{g_{t+1}\|w_t\|}{g_t\|w_{t+1}\|} - \eta \frac{g_{t+1}}{\|w_{t+1}\|} \langle \frac{w_t}{\|w_t\|^2}, A^T r \rangle)$$

$$(\text{rPGD}) \quad \lambda_t = (g_t\|v_t\| - g_{t+1})/(\eta_t g_t^2 g_{t+1}).$$

The regularization parameter $\lambda_t$ is highly dependent on $g_t$, $g_{t+1}$ and the input matrix $A$. However, it is difficult to characterize the behavior of $\lambda_t$ in general. In particular, we require the parameters $g_t$, $g_{t+1}$, $w_t$ and $w_{t+1}$ updated in a way that $\lambda_t > 0$. For the simpler setting of orthogonal $A$, we can see for rPGD that: 1) If the learning rate of $g$ is small enough, we will have $g_{t+1} < g_t\|v_t\|$, which means that $\lambda_t > 0$; 2) When $g_t w_t$ is close to $g^* w^*$, we will have $\|v_t\| \approx 1$, and $g_{t+1} \approx g_t$, which means that $\lambda_t \approx 0$.

# B  Proof of Lemma 2.2

*Proof.* **rPGD.** First we start with the reparametrized Projected Gradient Descent algorithm. The gradients for rPGD are

$$\nabla_w f(w, g) = g \nabla L(gw), \ \nabla_g f(w, g) = w^T \nabla L(gw),$$

First, the gradient step on $g$ clearly leads to the gradient flow for $g$. Second, for the update on $w$, we expand all terms to first order in $\eta$. Let $a_t = \nabla_{w_t} f(w_t, g_t)$. We start by expanding the squared Euclidean norm

$$\|w_t - \eta \nabla_{w_t} f(w_t, g_t)\|_2^2 = \|w_t - \eta a_t\|_2^2$$
$$= \|w_t\|^2 - 2\eta w_t^\top a_t + \eta^2 \|a_t\|_2^2$$
$$= 1 - 2\eta w_t^\top a_t + O(\eta^2).$$

On the last line, we have used that the iterates are normalized, so $\|w_t\|^2 = 1$.

Now, we can use the expansion $(1 + x)^{1/2} = 1 + x/2 + O(x^2)$, valid for $|x| \ll 1$, on the right hand side of the above display, to get

$$\|w_t - \eta \nabla_{w_t} f(w_t, g_t)\|_2 = 1 - \eta w_t^\top a_t + O(\eta^2).$$

Next we use this expansion in the update rule for the weights:

$$\begin{aligned}
w_{t+1} &= \frac{w_t - \eta \nabla_{w_t} f(w_t, g_t)}{\|w_t - \eta \nabla_{w_t} f(w_t, g_t)\|_2} \\
&= \frac{w_t - \eta a_t}{1 - \eta w_t^\top a_t + O(\eta^2)} \\
&= (w_t - \eta a_t) \cdot (1 + \eta w_t^\top a_t + O(\eta^2)).
\end{aligned}$$

In the last line, we have used the expansion

$$\frac{1}{1 - \eta x + O(\eta^2)} = 1 + \eta x + O(\eta^2)$$

valid for $|\eta| \ll 1$ and $x$ of a constant order. Recall now that for an arbitrary vector $w$, we defined $P_{w\perp} = I - \frac{ww^\top}{\|w\|_2^2}$ as the projection into the orthocomplement of $w$. Since $\|w_t\| = 1$, we have $\mathcal{P}_t = I - w_t w_t^\top$. By expanding the product and rearranging, keeping only the terms of larger order than $\eta$, we find

$$\begin{aligned}
w_{t+1} &= w_t - \eta a_t + \eta w_t w_t^\top a_t + O(\eta^2) \\
&= w_t - \eta \mathcal{P}_t a_t + O(\eta^2).
\end{aligned}$$

Taking $\eta \to 0$ and substituting the expression for $a_t$ and $\nabla_w f(w, g)$, we obtain the rPGD flow dynamics for $w$, i.e., $\dot{w}_t = -g_t \mathcal{P}_t \nabla L(g_t w_t)$. The update rule for $g$ follows directly.

**WN.** We now study weight normalization Salimans and Kingma [2016]. The WN objective function can be written using the loss function $L(x) = \|Ax - y\|^2/2$ as

$$h(g, w) = L\left(g \frac{w}{\|w\|}\right).$$

The discrete time algorithm is thus updated as

$$\begin{aligned}
v_t &= w_t/\|w_t\| \\
r_t &= y - g_t A v_t \\
g_{t+1} &= g_t - c\eta \cdot \left\langle v_t, \nabla L\left(g_t \frac{w_t}{\|w_t\|}\right)\right\rangle \\
w_{t+1} &= w_t - \eta \cdot g_t \cdot \mathcal{P}_t \frac{1}{\|w_t\|} \nabla L\left(g_t \frac{w_t}{\|w_t\|}\right).
\end{aligned}$$

When $\eta \to 0$, we recover the gradient flow on $g_t$ and $w_t$, i.e., recalling $v_t = w_t/\|w_t\|$

$$\begin{aligned}
\dot{g}_t &= -c \cdot \langle v_t, \nabla L(g_t v_t)\rangle \\
\dot{w}_t &= -g_t \cdot \mathcal{P}_t \frac{1}{\|w_t\|} \nabla L(g_t v_t) = -\frac{g_t}{\|w_t\|} \cdot \mathcal{P}_t \nabla L(g_t v_t).
\end{aligned}$$

Note the fact $\frac{d\|w_t\|^2}{dt} = 2 w_t^T \dot{w}_t = 0$ which gives $\|w_t\| = \|w_0\| = 1$.

Hence, for WN with initialization $\|w_0\| = 1$, we have that $g_t, v_t$ evolves exactly equivalent to the rPGD flow. The final formula for WN and rPGD that we will analyze is:

$$\begin{aligned}
\dot{g}_t &= -c \cdot w_t^T A^T r_t \\
\dot{w}_t &= -g_t \cdot \mathcal{P}_t A^T r_t.
\end{aligned}$$

$\square$

# C  Remaining Proofs for Section 2

## C.1  Proof of Lemma 2.3

*Proof.* At a stationary points of the loss, we have, with $r = y - Agw$

$$\partial_g h(w,g) = w^T A^T r = 0$$
$$\partial_w h(w,g) = g \cdot P_{w^\perp} A^T r = 0.$$

If $g \neq 0$, then we get $P_{w^\perp} A^T r = 0$. By adding this up with the first equation, we get $A^T r = 0$. Using that the smallest eigenvalue of $AA^T$ is nonzero, we conclude that $r = 0$. Hence this is stationary point with zero loss, which is also a global minimum.

Else, if $g = 0$, we see that $r = y - Agw = y$. Hence in this case, the stationary points belong to the set $S := \{(g,w) : g = 0, y^T Aw = 0\}$. This finishes the proof. $\qquad\square$

## C.2  Proof of Lemma 2.4 (dynamics of loss $\|r_t\|$)

We have

$$
\begin{aligned}
d[1/2\|r_t\|^2]/dt = r_t^T \dot{r}_t = r_t^T A d(g_t w_t)/dt \\
= r_t^T A[\dot{g}_t w_t + g_t \dot{w}_t] \\
= -r_t^T A[c \cdot w_t w_t^T A^T r_t + g_t^2 \mathcal{P}_t A^T r_t] \\
= -r_t^T A[c \cdot w_t w_t^T + g_t^2 \mathcal{P}_t] A^T r_t
\end{aligned}
$$

Thus,

$$d[1/2\|r_t\|^2]/dt \leq -\min\{g_t^2, c\}\|A^T r_t\|^2, \quad d[1/2\|r_t\|^2]/dt \geq -\max\{g_t^2, c\}\|A^T r_t\|^2. \quad (9)$$

We get a geometric convergence of the loss to zero, as soon as we can get a lower bound on $g_t^2$, which will be discussed below. If we have $g_t^2 \geq C^2$ for some constant $C^2 > 0$, we have

$$\min(g_t^2, c) r_t^T AA^T r_t \geq \min(C^2, c)\lambda_{\min}(AA^T)\|r_t\|^2$$

and so with $k := \min(C^2, c)\lambda_{\min}(AA^T)$,

$$d[1/2 \cdot \|r_t\|^2]/dt \leq -k\|r_t\|^2 \quad \Rightarrow \quad \|r_t\|^2 \leq \exp(-kt)\|r_0\|^2.$$

## C.3  Proof of Lemma 2.5

Define $P^\perp$ the projection into the orthocomplement of the row span of $A$, hence $w^\perp = P^\perp w$ and $P^\perp A^T = 0$. For simplicity, write $h(w_t, g_t) = h_t$

$$
\begin{aligned}
\frac{dw_t^\perp}{dt} = P^\perp \frac{dw_t}{dt} = -\frac{g_t}{\|w_t\|} P^\perp (I - \frac{w_t w_t^\top}{\|w_t\|^2}) \nabla_w h_t = \frac{g_t}{\|w_t\|} P^\perp \frac{w_t w_t^\top}{\|w_t\|^2} \nabla_w h_t = \frac{g_t}{\|w_t\|} P^\perp w_t (\nabla_g h_t) \\
= -\frac{g_t}{\|w_t\|^2} P^\perp w_t \left(\frac{1}{c}\frac{dg_t}{dt}\right) = -\frac{1}{2}\frac{P^\perp w_t}{\|w_t\|^2}\frac{dg_t^2}{c}{dt} = -\frac{1}{2}\frac{w_t^\perp}{\|w_0\|^2}\frac{dg_t^2}{c}{dt}.
\end{aligned}
$$

The last equality due to the fact that $d\|w_t\|^2/dt = 2w_t^T \dot{w}_t = 0$, which is also observed in [Tian et al., 2019, Tian, 2019]. Solving the dynamics $\frac{dw_t^\perp}{dt} = -\frac{1}{2}\frac{w_t^\perp}{c\|w_0\|^2}\frac{dg_t^2}{dt}$ with $\|w_0\| = 1$ results in (6).

## C.4  Proof of Theorem 2.6 (Convergence in the general case)

### C.4.1  Proof that either the loss converges to zero, or the iterates $(g_t, w_t)$ converge to the stationary set $\mathcal{S}$ defined in Lemma 2.3.

We start with the ODE for the loss,

$$d[1/2\|r_t\|^2]/dt = -r_t^T A[c \cdot w_t w_t^T + g_t^2 \mathcal{P}_t] A^T r_t. \quad (10)$$

This shows that the loss is decreasing, possibly not strictly.

If we have $g_t^2$ bounded away from zero, then the loss converges to zero geometrically. Thus, the only remaining case is when $g_t \to 0$.

Now, we have that the iterates take the form $x_t = g_t w_t$, and $\|w_t\| = 1$ are bounded. Hence, as $g_t \to 0$, we must have $x_t \to 0$.

Since the loss is continuous, we also have $\|r_t\|^2 = \|y - A g_t w_t\|^2 \to \|y\|^2$.

Suppose that the loss does not converge to zero. Since the loss is decreasing, this means that $\|r_t\| \to c > 0$ for some constant $c$.

From Equation (10), this implies that

$$(r_t^T A w_t)^2 \to 0.$$

Else, if this quantity is bounded away from zero, then $d[1/2\|r_t\|^2]/dt < -c'$ for some $c' > 0$, which would show $\|r_t\|$ decreases unboundedly, and is a contradiction.

Thus, we conclude that if the residual $r_t$ does not converge to zero, then the iterates $x_t = g_t w_t$ converge to zero: $x_t \to 0$. Moreover, $g_t \to 0$ and $y^T A w_t \to 0$. Given that $w_t$ is bounded, this shows that $(g_t, w_t)$ converges to the *set* of those stationary points of the loss characterized by

$$S := \{(g, w) : g = 0, y^T A w = 0\}.$$

Note specifically that we have not shown that $w_t$ converges to a specific stationary point, but rather only that it converges to the set $S$ of stationary points given above.

Now, in general, to avoid converging to this set, we must make some additional assumptions on the initialization. For instance, if we initialize with $g_0, w_0$ such that the loss is less than the loss at $x_0 = 0$, then, since we know that the loss is non-increasing, this shows that we will avoid converging to the above set of "bad" stationary points.

This result does not give a rate of convergence, so it is weaker than the result when $g_t$ is bounded away from zero. However, it is also more general. Initializing such that the loss is less than the loss at zero can be achieved conveniently, because we can calculate the value of the loss.

### C.4.2 Part I: Characterizing the solution when $c > 0$

The characterization follows by tracking the dynamics of the components in the row span of $A$ and its orthocomplement separately. The component in the row span converges to $x^*$. The normalized component $w_t^\perp$ in the orthocomplement is characterized by the invariant from the prior lemma. The scale of that component converges to $g^*$, as $g_t \to g^*$. This gives the desired result.

### C.4.3 Part II: Fixed $g_t$, i.e. $c = 0$

For the fixed $g_t$ case, we have

$$\begin{aligned} d[1/2 \cdot \|r_t\|^2]/dt = r_t^T \dot{r}_t &= r_t^T A d(g_t w_t)/dt \\ &= r_t^T A[\dot{g}_t w_t + g_t \dot{w}_t] \\ &= r_t^T A g_0 \dot{w}_t \\ &= -r_t^T A g_0 \mathcal{P}_t g_0 A^T r_t \\ &= -g_0^2 \|\mathcal{P}_t A^T r_t\|^2. \end{aligned}$$

Now, it follows that $\|r_t\|$ is a non-increasing quantity, which is also strictly decreasing as long as $\|\mathcal{P}_t A^T r_t\| > 0$. It also follows that as $t \to \infty$, we have $\|\mathcal{P}_t A^T r_t\| \to 0$. Now, since $g_0 < g^*$, it follows that $A^T r_t$ has a norm that is strictly bounded away from zero, i.e., $\|A^T r_t\| > c_0 > 0$ for some $c_0 > 0$. So, we do not have the residual going to zero in this case. Hence, this can also be written as $w_t = c_t \cdot A^T r_t / \|A^T r_t\|$, for some sequence of scalars $c_t$ with $\liminf c_t^2 > 0$.

Hence, $w_t$ becomes asymptotically parallel to the row space of $A$. Now, since $w_t$ lives on the compact space of unit vectors, considering any subsequence of it, it also follows that it has a convergent subsequence. Let $w$ be the limit along any convergent subsequence. It follows that $w = \pm A^T r / \|A^T r\|$. Next we note that the solution with $+$ actually *maximizes* the loss over

$\|w\| = 1$. Hence, the only possible solution is $w = -A^T r / \|A^T r\|$. Since this holds for any convergent subsequence, it follows that $w_t$ itself must converge.

Now we get a more explicit form for the solution $w$. We can say that $w$ is the unique unit norm vector such that $w = -\hat{c} A^T r$, for some $\hat{c} > 0$. Then we can write that equation as

$$w = -\hat{c} A^T (Agw - y)$$
$$(I + cgA^T A)w = \hat{c} A^T y$$
$$w = (\hat{c}^{-1} I + g A^T A)^{-1} A^T y.$$

Thus, $w$ is the unique vector of the above form such that $\|w\| = 1$. This can be viewed as a form of implicit regularization. Namely, $w$ is the unique vector, for which there is some regularization parameter $\hat{c}$ such that, $w$ minimizes the regularized least squares objective

$$\frac{1}{2} \|Agw - y\|^2 + \frac{g}{2\hat{c}} \|w\|^2$$

and $\|w\| = 1$. This $w$ will in general not be the pointing in the direction of the optimal solution. We recall that the optimal solution $w^*$ has the form $w^* = A^\dagger b / \|A^\dagger y\|$, where $A^\dagger$ is the pseudoinverse of $A$. We recall that the action of the pseudoinverse can be characterized as the limit of ridge regularization with infinitely small penalization, i.e., $A^\dagger y = \lim_{\lambda \to 0} (A^T A + \lambda I)^{-1} A^T y$. For orthogonal $A$, we see that $w$ converges to the right direction, because $A^T = A^\dagger$. However, for general $A$, the flow does not in general converge to the min-norm direction.

Now, suppose we start the flow for both $g_t, w_t$ again from a point $w_0$ that belongs to the span of the row space of $A$ (call it $R$). Then, by the update rule for $w_t$, it follows that $w_t \in R$ for all $t$. Therefore, the derivative of the loss becomes

$$d[1/2 \cdot \|r_t\|^2]/dt = r_t^T \dot{r}_t = r_t^T A d(g_t w_t)/dt$$
$$= r_t^T A[\dot{g}_t w_t + g_t \dot{w}_t]$$
$$= -r_t^T A[w_t w_t^T A^T r_t + g_t^2 \mathcal{P}_t A^T r_t]$$
$$= -(r_t^T A w_t)^2.$$

From arguments similar to before, it follows that $r_t^T A w_t \to 0$ as $t \to \infty$. Moreover, from a similar subsequence argument it also follows that $w_t \to w$ such that $r^T A w = 0$. Since $\|w\| = 1$ and $A$ has full row rank, it follows that $r = 0$. Hence the flow converges to a zero of the loss. Moreover, since $w \in R$, it follows that this is the minimum norm solution.

### C.5  Proof of Theorem 2.7 (Convergence rate)

#### C.5.1  First case

Note first that by (6), we have

$$g_t^2 = 2 \log \|w_0^\perp\| + g_0^2 - 2 \log \|w_t^\perp\| \geq 2 \log \|w_0^\perp\| + g_0^2 \tag{11}$$

where the last inequality is due to the fact $\|w_t^\perp\| \leq \|w_t\| = 1$. Thus, we have our first lower bound. From Lemma 2.4, we get the convergence rate for the first case.

#### C.5.2  Second case

**Lemma C.1** (Lower bound on $g_t$). *Under the same setting of Lemma 2.2, suppose the initialization of $g_0$ and $w_0$ satisfies $\langle Aw_0, Aw^* \rangle > 0$ and $g_0 \geq 0$. Let $\frac{\|Ag^* w^*\|^2 - \|A(g_0 w_0 - g^* w^*)\|^2}{\lambda_{\max}} > \delta$ for some small $\delta$.* [4] *We have the lower bound of $g_t$:*

$$g_t \geq \frac{\|Ag^* w^*\|^2 - \|A(g_0 w_0 - g^* w^*)\|^2)}{\lambda_{\max}} - \frac{\delta}{2} \text{ for } t \geq s$$

*where the time $s$ depends on our initialization $g_0 > 0$:*

$$s = \begin{cases} 0 & \text{if } g_0 \geq \min\left\{ \frac{2g^*\langle Aw_0, Aw^*\rangle}{\|Aw_0\|^2}, \frac{\|Ag^*w^*\|^2 - \|A(g_0 w_0 - g^*w^*)\|^2)}{\lambda_{\max}} - \delta \right\}, \\ \frac{1}{\lambda_{\max}} \log\left( \frac{2}{\delta}\left( \frac{\|Ag^*w^*\|^2 - \|A(g_0 w_0 - g^*w^*)\|^2}{\lambda_{\max}} - g_0 \right) \right) & \text{otherwise.} \end{cases}$$

**Proof of Lemma C.1.** We start with the dynamics

$$\frac{dg_t}{dt} = -\langle Aw_t, A(g_t w_t - g^*w^*)\rangle = -g_t\|Aw_t\|^2 + g^*\langle Aw_t, Aw^*\rangle$$

$$\frac{dg_t^2}{dt} = -2\langle Ag_t w_t, A(g_t w_t - g^*w^*)\rangle = \|Ag^*w^*\|^2 - \|A(g_t w_t - g^*w^*)\|^2 - \|Ag_t w_t\|^2$$

There is a geometric interpretation. Consider the points $0$, $z^* = Ag^*w^*$, and $z_t = Ag_t w_t$. Then, we have

$$\frac{dg_t^2}{dt} = \|z^*\|^2 - \|z_t - z^*\|^2 - \|z_t\|^2.$$

By the Pythagorean theorem, this quantity is positive if and only if the angle between the vectors $z_t$ and $z^* - z_t$ is "acute" (i.e., less than $\pi/2$ radians, or 90 degrees). This can be visualized by drawing the triangle with vertices at $0$, $z^*$, and $z_t$, and noting that the angle in question is the one at the vertex $z_t$. Thus, the above result states that as long as that angle is acute, $g_t$ will increase. The general convergence result states that $\|z_t - z^*\|^2$ is non-increasing, while our initalization condition states that $\|z^*\|^2 - \|z_0 - z^*\|^2 > 0$. Moreover, $\|z_t\|^2$ can be upper bounded by a constant times $g_t^2$. Taken together, these facts imply that if $g_t^2$ becomes small, then $\frac{dg_t^2}{dt}$ becomes positive, and hence $g_t^2$ starts to grow again. Thus, the dynamics of $g_t^2$ show a "self-balancing" dynamics. This is made rigorous below.

Denote $\delta_t = \frac{2g^*\langle Aw_t, Aw^*\rangle}{\|Aw_t\|^2}$, then

$$\frac{dg_t}{dt} = \|Aw_t\|^2 (\delta_t - g_t) \quad \text{and} \quad \frac{dg_t^2}{dt} = g_t\|Aw_t\|^2 (\delta_t - 2g_t) \tag{12}$$

$$\|Ag^*w^*\|^2 - \|A(g_t w_t - g^*w^*)\|^2 = g_t\|Aw_t\|^2 \left( \frac{2g^*\langle Aw_t, Aw^*\rangle}{\|Aw_t\|^2} - g_t \right) = g_t\|Aw_t\|^2 (\delta_t - g_t)$$

From this condition: $\|Ag^*w^*\|^2 - \|A(g_0 w_0 - g^*w^*)\|^2 = \|Aw_0\|^2 (g_0\delta_0 - g_0^2) > 0$, we have that $\langle Ag_0 w_0, Ag^*w^*\rangle > 0$. Wihtout loss of generality, we assume $g_t > 0$ and $\langle Aw_0, Ag^*w^*\rangle > 0$. [5]

$$\delta < g_0 < \min\left\{ \delta_0, \frac{(\delta_0 - g_0)}{\lambda_{\max}} g_0\|Aw_0\|^2 \right\} \tag{13}$$

which means

$$g_0 < \frac{2g^*\langle Aw_0, Aw^*\rangle}{\|Aw_0\|^2} \quad \Leftrightarrow \quad \|Ag^*w^*\|^2 > \|A(g_0 w_0 - g^*w^*)\|^2$$

$$g_0 < \frac{(\delta_0 - g_0)}{\lambda_{\max}} g_0\|Aw_0\|^2 \quad \overset{(12)}{\Leftrightarrow} \quad \|Ag^*w^*\|^2 - \|A(g_0 w_0 - g^*w^*)\|^2 - \lambda_{\max}g_0^2 > 0.$$

Since we always have $\|A(g_t w_t - g^*w^*)\| \geq \|A(g_s w_s - g^*w^*)\|$ for $t > s$, it implies

$$\frac{dg_t^2}{dt} \geq \|Ag^*w^*\|^2 - \|A(g_0 w_0 - g^*w^*)\|^2 - \lambda_{\max}g_t^2$$

$$= \lambda_{\max}\left( \frac{(\delta - g_0)}{\lambda_{\max}} g_0\|Aw_0\|^2 - g_t^2 \right) > 0 \tag{14}$$

Solving (14) gives

$$g_t^2 \geq \left( g_0 - \frac{(\delta - g_0)}{\lambda_{\max}} g_0\|Aw_0\|^2 \right) e^{-\lambda_{\max}t} + \frac{(\delta - g_0)}{\lambda_{\max}} g_0\|Aw_0\|^2$$

This means that for initialization $g_0$ satisfying (13), it will take time at most

$$s = \frac{1}{\lambda_{\max}} \log \left( \frac{2}{\delta} \left( \frac{(\delta - g_0)}{\lambda_{\max}} g_0 \|Aw_0\|^2 - g_0 \right) \right)$$

such that $g_s \geq \frac{(\delta - g_0)}{\lambda_{\max}} g_0 \|Aw_0\|^2 - \frac{1}{2}\delta$. Thus after $s$,

$$\inf_{t \geq s} g_t \geq \frac{(\delta - g_0)}{\lambda_{\max}} g_0 \|Aw_0\|^2 - \frac{1}{2}\delta.$$

## D Beyond Linear Regression

Here we illustrate that the invariant in the optimization path holds more generally than for linear regression, and specifically for certain general loss functions that only depend on a small dimensional subspace of the parameter space. Let $L : \mathbb{R}^d \to \mathbb{R}$ be the loss function, and our goal is to solve

$$\min_{x \in \mathbb{R}^d} L(x) \tag{15}$$

where $L(x)$ is differentiable and satisfies Assumption D.1.

**Assumption D.1** (Low-dimensional gradient). *There exists a projection matrix $P \in \mathbb{R}^{d \times d}$ with rank $r < d$ such that*

$$(I - P)\nabla L(x) = 0, \forall x \in \mathbb{R}^d.$$

Let $P^\perp = (I - P)$. Assumption D.1 is equivalent to the fact that the gradient of $L$ lives in the low-dimensional space given by the span of $P$, $\nabla L(x) \in span(P)$. This implies

$$L(x) = L(Px) \quad \forall x \in \mathbb{R}^d.$$

This means that the objective only depends on the projection of $x$ into the span of $P$. To use the orthogonal projection in what follows, define $x^{\|} = Px$ and $x^\perp = P^\perp x$. For the undetermined linear regression, $P = A^\dagger$ where $A^\dagger$ is the pseudo-inverse of the matrix $A$.

**Theorem D.2** (WN flow Invariance for General Loss). *Consider the loss function in (15) with Assumption D.1. The WN method transforms the loss function to $h(g, w) = L\left(g_t \frac{w_t}{\|w_t\|}\right)$. The WN gradient flow from Algorithm 1 with initial condition $(w_0, g_0)$, started from $w_0$ with not necessarily unit norm, has the invariant*

$$w_t^\perp = \exp\left(\frac{g_0^2 - g_t^2}{2\|w_0\|^2}\right) w_0^\perp.$$

The proof of the above theorem is a simple extension of the proof of Lemma2.5 with $A^\top r_t = \nabla L\left(g_t \frac{w_t}{\|w_t\|}\right)$. This result suggests that the reason for the invariance is that the original objective function before reparametrization only depends on a smaller dimensional space.

## E Remaining Proofs for Section 3

### E.1 Proof of Theorem 3.2 Case (a)

We restate the case (a) of Theorem 3.2 here.

**Theorem E.1** (Updating $g$ in Phase I). *Suppose we initialize with $g_0 < g^*$. Let $\delta_0 = (g^*)^2 - (g_0)^2 > 0$, $0 < \rho < 1$ and $\delta < \frac{\varepsilon}{2g^* + \varepsilon}$. Suppose the number of iterations $T_1$ and $T_2$ is of the order:*

$$T_1 = \left(1 + \frac{(g^*)^2}{\rho \delta_0}\right) \log \left(\frac{1 - \|Aw_0\|^2}{\delta^2}\right) = \mathcal{O}\left(\frac{(g^*)^2}{\rho \delta_0} \log \left(\frac{1}{\delta^2}\right)\right),$$

$$T_2 = \frac{1}{\gamma^{(2)}} \log \left(\frac{\rho \delta_0}{g^{*2}(1 - \varepsilon)\sqrt{\varepsilon}}\right) = \mathcal{O}\left(\frac{1}{\gamma^{(2)}} \log \left(\frac{\rho(g^{*2} - g_0^2)}{g^{*2}\sqrt{\varepsilon}}\right)\right).$$

*Fix $g_1 = g_0$ at the first step. For iterations $t = 1, \ldots, T_1 - 1$, set the stepsize for $g$ to*

$$\gamma^{(1)} < \min \left\{ \frac{\|Ag_0w_0\|^2}{2T_1((g^*)^2 - \|Ag_0w_0\|^2)} \log \left((1 - \rho)\frac{g^{*2}}{g_0^2} + \rho\right), \frac{1}{2(1 + \|Aw_0\|^2)} \right\}$$

and to any $\gamma^{(2)} \leq \frac{1}{4}$ for $t = T_1, T_1 + 1, \ldots, T_1 + T_2 - 1$. Set $\eta_t = 1/g_t^2$. Then we reach $\|w_T^\perp\| \leq \varepsilon$ and $\|Ag_T w_T - b\|^2 \leq \varepsilon g^{*2}$ after $T = T_1 + T_2$ iterations.

*Proof.* **The norm** $\|w^\perp\|$. Let us first get the upper bound of $g_{t+1}$ to see how the norm of $\|w_t^\perp\| = 1 - \|Aw_t\|^2$ evolves. Suppose that we have for some $0 < \rho < 1$

$$(g^*)^2 - \|Ag_{T_1} w_{T_1}\|^2 \geq (g^*)^2 - g_{T_1}^2 = \rho \delta_0 \tag{16}$$

By Lemma E.6, we have

$$\|w_{T_1}^\perp\|^2 = (1 - \|Aw_{T_1}\|^2) \leq \exp\left(-\sum_{i=1}^{T_1} \frac{(g^*)^2 - \|Ag_i w_i\|^2}{(g^*)^2 + (g^*)^2 - \|Ag_i w_i\|^2}\right)(1 - \|Aw_0\|^2) \tag{17}$$

$$\leq \exp\left(-\sum_{i=1}^{T_1} \frac{(g^*)^2 - \|Ag_{T_1} w_{T_1}\|^2}{(g^*)^2 + (g^*)^2 - \|Ag_{T_1} w_{T_1}\|^2}\right)(1 - \|Aw_0\|^2) \tag{18}$$

$$\leq \exp\left(-\frac{\rho \delta_0 T_1}{(g^*)^2 + \rho \delta_0}\right)(1 - \|Aw_0\|^2) \tag{19}$$

we have $\|w_{T_1}^\perp\|^2 = 1 - \|Aw_{T_1}\|^2 \leq \delta^2$ when

$$T_1 = \left(1 + \frac{(g^*)^2}{\rho \delta_0}\right) \log\left(\frac{1 - \|Aw_0\|^2}{\delta^2}\right).$$

Now we only need to verify condition (16). To see this, notice that we use Lemma E.3:

$$g_{t+1} \leq g_t + \frac{\gamma g_t}{2}(\|v_t\|^2 - 1) \text{ for } t < T_1$$

Note $\|v_t\|^2 = g^{*2}/(g_t^2 \|Aw_{t+1}\|^2) > 1$ as $\|Aw_{t+1}\| < 1$ and $g^{*2} > g_t^2$. Thus, even though $g_t$ grows with the rate $\gamma^{(1)}(\|v_{t-1}\|^2 - 1)$, we use our choice of $\gamma^{(1)}$ and $g_{t+1} < g^*$. In fact, we set $\gamma^{(1)}$ small such that after $T_1$ there is a gap between $g^*$ and $g_{T_1}$. We let the gap satisfies $(g^*)^2 - g_{T_1}^2 \geq \rho \delta_0$:[6]

$$g_{T_1+1}^2 \leq \prod_{t=1}^{T_1+1}\left(1 + \frac{1}{2}\gamma^{(1)}(\|v_{t-1}\|^2 - 1)\right)^2 g_1^2 \overset{(a)}{\leq} \exp\left(\gamma^{(1)}(\|v_0\|^2 - 1)T_1\right) g_0^2 \overset{(b)}{\leq} (g^*)^2 - \rho \delta_0 \tag{20}$$

where step $(a)$ due to $g_1 = g_0$ and the that

$$\frac{\|v_t\|^2}{\|v_{t-1}\|^2} \leq \frac{g_{t-1}^2}{g_t^2} \leq 1 \tag{21}$$

which is due to $g_t^2 \|v_t\|^2 - g_{t-1}^2 \|v_{t-1}\|^2 < 0$ (Lemma E.5 and $g_t - g_{t-1}\|v_{t-1}\| \leq g^* - \frac{g^*}{\|Aw_t\|} < 0$) In step $(b)$ as long as we make sure

$$\gamma^{(1)} \leq \frac{\log\left((g^{*2} - \rho\delta_0)/g_0^2\right)}{(\|v_0\|^2 - 1)T_1} = \frac{\|Ag_0 w_0\|^2}{2T_1((g^*)^2 - \|Ag_0 w_0\|^2)} \log\left((1-\rho)\frac{g^{*2}}{g_0^2} + \rho\right)$$

which is satisfied by our choice of $\gamma^{(1)}$ for fixed $T_1$, $g_0$, and $\delta_0$.

**The loss** $\|A(g_T w_T - g^* w^*)\|$. By Lemma E.2,

$$\|A(g_{t+1} w_{t+1} - g^* w^*)\|^2 = (g_{t+1} - g_t\|v_t\|)^2$$

which means we only need to analyze the term

$$g_t\|v_t\| - g_{t+1} = (1 - \gamma^{(1)})(g_{t-1}\|v_{t-1}\| - g_t) \tag{22}$$

$$- (g_{t-1}\|v_{t-1}\| - g_t\|v_t\|)\left(1 - \frac{\gamma^{(1)}}{g_t}\frac{g_{t-1}\|v_{t-1}\|}{(g_t + g_{t-1}\|v_{t-1}\|)}(g_t\|v_t\| + g_{t-1}\|v_{t-1}\|)\right).$$

Again, for $t < T_1$, we have from (21) that $g_{t-1}\|v_{t-1}\| - g_t\|v_t\| > 0$. Meanwhile, with Lemma E.4 and Lemma E.5, we have

$$\frac{g_t(g_t + g_{t-1}\|v_{t-1}\|)}{g_{t-1}\|v_{t-1}\|(g_t\|v_t\| + g_{t-1}\|v_{t-1}\|)} \geq \frac{g_t^2}{g_{t-1}\|v_{t-1}\|(g_t\|v_t\| + g_{t-1}\|v_{t-1}\|)}$$

$$\geq \frac{g_0^2}{2g_0^2\|v_0\|^2}$$

$$\geq \frac{1}{2(1 + \|Aw_0\|^2)}$$

By our choice of $\gamma^{(1)}$,

$$\gamma^{(1)} \leq \frac{1}{2\|v_0\|^2} \leq \frac{g_t(g_t + g_{t-1}\|v_{t-1}\|)}{g_{t-1}\|v_{t-1}\|(g_t\|v_t\| + g_{t-1}\|v_{t-1}\|)}$$

We have from (22) that $t > T_1$

$$g_t\|v_t\| - g_{t+1} \leq (1 - \gamma^{(2)})^{t-T_1}(g_{T_1}\|v_{T_1}\| - g_{T_1})$$

$$= (1 - \gamma^{(2)})^{t-T_1}(g_{T_1}^2\|v_{T_1}\|^2 - g_{T_1}^2)/(g_{T_1}\|v_{T_1}\| + g_{T_1})$$

$$= (1 - \gamma^{(2)})^{t-T_1}\left(g^{*2}/\|Aw_{T_1}\|^2 - g_{T_1}^2\right)/(g^*/\|Aw_{T_1}\| + g_{T_1}) \tag{23}$$

$$\leq (1 - \gamma^{(2)})^{t-T_1}\left(g^{*2} - g_{T_1}^2\right)/(g^*(1 - \varepsilon) + g_{T_1}) \tag{24}$$

$$= (1 - \gamma^{(2)})^{t-T_1}\rho\delta_0/(g^*(1 - \varepsilon) + g_{T_1}) \tag{25}$$

$$\leq (1 - \gamma^{(2)})^{t-T_1}\rho\delta_0/(g^*(1 - \varepsilon)). \tag{26}$$

So we have $g_T\|v_T\| - g_T \leq g^*\sqrt{\varepsilon}$ after $T = T_1 + \frac{1}{\gamma^{(2)}}\log\left(\frac{\rho\delta_0}{(g^{*2}(1-\varepsilon))\sqrt{\varepsilon}}\right)$

$\square$

### E.1.1 Technical Lemmas for Theorem E.1

In the following section, we assume $AA^\top = I_{m \times m}$ and use $r_t = A(g_t w_t - g^* w^*)$ to denote the negative residual.

**Lemma E.2.** *With the step-size $\eta = \frac{1}{g_t^2}$, we have the following equalities: We have the following property:*

$$Aw_{t+1} = \frac{Ag^*w^*}{g_t\|v_t\|} \tag{27}$$

$$g_t^2\|v_t\|^2 = g_t^2 + ((g^*)^2 - \|Ag_t w_t\|^2) = \frac{g^{*2}}{\|Aw_{t+1}\|^2} \tag{28}$$

$$\|Aw_{t+1}\|^2 = \frac{(g^*)^2}{(g^*)^2 + g_t^2(1 - \|Aw_t\|^2)} \tag{29}$$

$$1 - \|Aw_{t+1}\|^2 = \frac{1}{\|v_t\|^2}\left(1 - \|Aw_t\|^2\right) \tag{30}$$

$$g_{t+1}^2 - \|Ag_{t+1}w_{t+1}\|^2 = \frac{\|Ag_{t+1}w_{t+1}\|^2}{(g^*)^2}\left(g_t^2 - \|Ag_t w_t\|^2\right). \tag{31}$$

*Proof.* With the update of $w_t$:

$$v_{t+1} = w_t - \eta g_t A^\top r_t, \qquad w_{t+1} = \frac{v_t}{\|v_t\|},$$

we can get

$$Aw_{t+1} = \frac{Aw_t - \frac{1}{g_t}r_t}{\|v_t\|} = \frac{Ag^*w^*}{g_t\|v_t\|},$$

and

$$\|v_t\|^2 = \|w_t\|^2 - 2\eta\langle Ag_tw_t, r_t\rangle + \eta^2 g_t^2 \|r_t\|^2$$
$$= 1 + \eta(\|Ag^*w^*\|^2 - \|Ag_tw_t\|^2 - \|r_t\|^2) + \eta^2 g_t^2 \|r_t\|^2$$

Moreover,

$$\|Aw_{t+1}\|^2 = \frac{1}{\|v_t\|^2}\|Aw_t - \eta g_t r_t\|^2$$
$$\Leftrightarrow \|v_t\|^2 \|Aw_{t+1}\|^2 = \|Aw_t\|^2 - 2\eta\langle Ag_tw_t, r_t\rangle + \eta^2 g_t^2\|r_t\|^2$$
$$= \|Aw_t\|^2 - 2\eta\langle Ag_tw_t, r_t\rangle + \eta^2 g_t^2\|r_t\|^2$$
$$= \|Aw_t\|^2 + \eta(\|Ag^*w^*\|^2 - \|Ag_tw_t\|^2 - \|r_t\|^2) + \eta^2 g_t^2\|r_t\|^2.$$

Letting $\eta = \frac{1}{g_t^2}$, we have:

$$g_t^2\|v_t\|^2 = g_t^2 + ((g^*)^2 - \|Ag_tw_t\|^2)$$

$$\|v_t\|^2\|Aw_{t+1}\|^2 = \frac{(g^*)^2}{g_t^2}.$$

From (28), we can get (29), and with (29), we can obtain (31) and (30) after some algebra. $\qquad\square$

**Lemma E.3.** *For $\eta = \frac{1}{g_t^2}$ and $\gamma = \gamma_t$, we have $g_{t+1} \leq g_t + \frac{\gamma g_t}{2}(\|v_t\|^2 - 1)$ and*

$$g_t\|v_t\| - g_{t+1} = (1 - \gamma)(g_{t-1}\|v_{t-1}\| - g_t)$$
$$- (g_{t-1}\|v_{t-1}\| - g_t\|v_t\|)\left(1 - \frac{\gamma}{g_t}\frac{g_{t-1}\|v_{t-1}\|}{(g_t + g_{t-1}\|v_{t-1}\|)}(g_t\|v_t\| + g_{t-1}\|v_{t-1}\|)\right)$$

*Proof.* The update of $g_{t+1}$ is

$$g_{t+1} = g_t - \gamma\langle Aw_t, r_t\rangle$$
$$= g_t + \frac{\gamma}{2g_t}((g^*)^2 - \|Ag_tw_t\|^2 - \|r_t\|^2)$$
$$= g_t + \frac{\gamma g_t}{2}(\|v_t\|^2 - 1) - \frac{\gamma}{2g_t}\|r_t\|^2.$$

where the second equality due to $\langle Ag_tw_t, r_t\rangle = \|Ag^*w^*\|^2 - \|Ag_tw_t\|^2 - \|r_t\|^2$ and the last equality due to update of $\|v_t\|$ (see equality 28). This finishs the proof for the first inequality.

Denoting $C_t = \alpha_{t-1}^2 g_{t-1}^2(\|Aw_{t-1}\|^2 - 1)$, we get

$$g_{t+1} = g_t + \frac{\gamma}{2g_t}g_t^2(\|v_t\|^2 - 1) - \frac{\gamma}{2g_t}\|r_t\|^2$$
$$= g_t + \frac{\gamma g_t}{2}\|v_t\|^2 - \frac{\gamma g_t}{2} - \frac{\gamma}{2g_t}\left((g_{t-1}\|v_{t-1}\| - g_t)^2 + C_t\right)$$
$$= g_t + \frac{\gamma g_t}{2}\|v_t\|^2 - \frac{\gamma g_t}{2} - \frac{\gamma}{2g_t}\left((g_{t-1}^2\|v_{t-1}\|^2 + g_t^2 - 2g_t g_{t-1}\|v_{t-1}\|) + C_t\right)$$
$$= g_t + \frac{\gamma g_t}{2}\|v_t\|^2 - \frac{\gamma}{2g_t}g_{t-1}^2\|v_{t-1}\|^2 + \gamma(g_{t-1}\|v_{t-1}\| - g_t) - \frac{\gamma}{2g_t}C_t$$
$$= g_t + \frac{\gamma}{2g_t}\left(g_t^2\|v_t\|^2 - g_{t-1}^2\|v_{t-1}\|^2\right) + \gamma(g_{t-1}\|v_{t-1}\| - g_t) - \frac{\gamma}{2g_t}C_t$$
$$\Rightarrow g_t\|v_t\| - g_{t+1} = g_t\|v_t\| - g_t - \frac{\gamma}{2g_t}\left(g_t^2\|v_t\|^2 - g_{t-1}^2\|v_{t-1}\|^2\right) - \gamma(g_{t-1}\|v_{t-1}\| - g_t) + \frac{\gamma}{2g_t}C_t$$
$$= (1 - \gamma)(g_{t-1}\|v_{t-1}\| - g_t) - g_{t-1}\|v_{t-1}\| + g_t\|v_t\|$$
$$\underbrace{- \frac{\gamma}{2g_t}\left(g_t^2\|v_t\|^2 - g_{t-1}^2\|v_{t-1}\|^2\right) + \frac{\gamma}{2g_t}\alpha_{t-1}^2 g_{t-1}^2(\|Aw_{t-1}\|^2 - 1)}_{Term1}.$$

We prove the lemma with following simplification for $Term1$:

$$
\begin{aligned}
Term1 &= -\frac{\gamma}{2g_t}\left(g_t^2\|v_t\|^2 - g_{t-1}^2\|v_{t-1}\|^2\right) + \frac{\gamma}{2g_t}\alpha_{t-1}^2 g_{t-1}^2(\|Aw_{t-1}\|^2 - 1) \\
&= -\frac{\gamma}{2g_t}\left(\frac{g_t^2}{\|v_{t-1}\|^2} - g_{t-1}^2\right)(1 - \|Aw_{t-1}\|^2) + \frac{\gamma}{2g_t}\alpha_{t-1}^2 g_{t-1}^2(\|Aw_{t-1}\|^2 - 1) \\
&= -\frac{\gamma}{2g_t}(1 - \|Aw_{t-1}\|^2)\left(\frac{g_t^2}{\|v_{t-1}\|^2} - g_{t-1}^2 - \left(\frac{g_t}{\|v_{t-1}\|} - g_{t-1}\right)^2\right) \\
&= -\frac{\gamma}{2g_t}(1 - \|Aw_{t-1}\|^2)\left(\frac{2g_t g_{t-1}}{\|v_{t-1}\|} - 2g_{t-1}^2\right) \\
&= -\frac{\gamma g_{t-1}}{g_t}(1 - \|Aw_{t-1}\|^2)\left(\frac{g_t}{\|v_{t-1}\|} - g_{t-1}\right) \\
&= -\frac{\gamma g_{t-1}\|v_{t-1}\|}{g_t}\frac{g_t^2\|v_t\|^2 - g_{t-1}^2\|v_{t-1}\|^2}{g_t + g_{t-1}\|v_{t-1}\|}
\end{aligned}
$$

where at the last step we use Lemma E.5 to have:

$$
\begin{aligned}
(1 - \|Aw_{t-1}\|^2)\left(\frac{g_t}{\|v_{t-1}\|} - g_{t-1}\right) &= \frac{1}{\frac{g_t}{\|v_{t-1}\|} + g_{t-1}}(1 - \|Aw_{t-1}\|^2)\left(\frac{g_t^2}{\|v_{t-1}\|^2} - g_{t-1}^2\right) \\
&= \frac{g_t^2\|v_t\|^2 - g_{t-1}^2\|v_{t-1}\|^2}{\frac{g_t}{\|v_{t-1}\|} + g_{t-1}} \\
&= \|v_{t-1}\|\frac{g_t^2\|v_t\|^2 - g_{t-1}^2\|v_{t-1}\|^2}{g_t + g_{t-1}\|v_{t-1}\|}
\end{aligned}
$$

$\square$

**Lemma E.4.** *If $\eta = \frac{1}{g_t^2}$ and $g_t - g_{t-1}\|v_{t-1}\| < 0$, we always have that*

$$g_{t+1} > g_t, \quad \forall t$$

*Proof.* Notice that $g_{t+1} = g_t - \gamma\langle Aw_t, r_t\rangle$, so we only need to prove that $\langle Aw_t, r_t\rangle < 0$. Indeed,

$$
\langle Aw_t, r_t\rangle = g_t\|Aw_t\|^2 - g^*\langle Aw_t, Aw^*\rangle = \frac{g_t(g^*)^2}{g_{t-1}^2\|v_{t-1}\|^2} - \frac{(g^*)^2}{g_{t-1}\|v_{t-1}\|} = \frac{(g^*)^2(g_t - g_{t-1}\|v_{t-1}\|)}{g_{t-1}^2\|v_{t-1}\|^2} < 0.
$$

$\square$

**Lemma E.5.** *We have the following identity for the recursion on $g_t^2\|v_t\|^2$:*

$$
g_t^2\|v_t\|^2 - g_{t-1}^2\|v_{t-1}\|^2 = \left(\frac{g_t^2}{\|v_{t-1}\|^2} - g_{t-1}^2\right)(1 - \|Aw_{t-1}\|^2) = (g_t^2 - g_{t-1}^2\|v_{t-1}\|^2)(1 - \|Aw_t\|)^2.
$$

*Proof.* By Lemma E.2, we use the (28) to get

$$
\begin{aligned}
g_t^2\|v_t\|^2 - g_{t-1}^2\|v_{t-1}\|^2 &= g_t^2(1 - \|Aw_t\|^2) - g_{t-1}^2(1 - \|Aw_{t-1}\|^2) \\
&= \left(\frac{g_t^2}{\|v_{t-1}\|^2} - g_{t-1}^2\right)(1 - \|Aw_{t-1}\|^2) \\
&= (g_t^2 - g_{t-1}^2\|v_{t-1}\|^2)(1 - \|Aw_t\|)^2.
\end{aligned}
$$

$\square$

**Lemma E.6.** *We have the following bound on the closeness of $Aw_t$ to unit norm:*

$$
\|w_t^\perp\| \leq (1 - \|Aw_t\|^2) \leq \exp\left(-\sum_{i=1}^{t}\frac{(g^*)^2 - \|Ag_iw_i\|^2}{(g^*)^2 + (g^*)^2 - \|Ag_iw_i\|^2}\right)(1 - \|Aw_0\|^2). \tag{32}
$$

*Proof.* If we keep $g_t \leq g^*$, by (29) we always have that

$$1 - \|Aw_{t+1}\|^2 = \frac{g_t^2(1 - \|Aw_t\|^2)}{(g^*)^2 + g_t^2(1 - \|Aw_t\|^2)}$$

$$\leq \frac{(g^*)^2}{(g^*)^2 + (g^*)^2 - \|Ag_t w_t\|^2}(1 - \|Aw_t\|^2)$$

$$\leq \exp(-\frac{(g^*)^2 - \|Ag_t w_t\|^2}{(g^*)^2 + (g^*)^2 - \|Ag_t w_t\|^2})(1 - \|Aw_0\|^2).$$

The first inequality holds due to

$$\frac{g_t^2}{(g^*)^2 + g_t^2(1 - \|Aw_t\|^2)} \leq \frac{g_t^2 + ((g^*)^2 - g_t^2)}{(g^*)^2 + g_t^2(1 - \|Aw_t\|^2) + ((g^*)^2 - g_t^2)}$$

Thus,

$$(1 - \|Aw_t\|^2) \leq \exp(-\sum_{i=1}^{t} \frac{(g^*)^2 - \|Ag_i w_i\|^2}{(g^*)^2 + (g^*)^2 - \|Ag_i w_i\|^2})(1 - \|Aw_0\|^2).$$

$\square$

## E.2 Proof of Theorem E.7 Case (b)

Here we discuss the case (b) of Theorem 3.2

**Theorem E.7** (Fixing $g$ in Phase I). *Suppose the initialization satisfies $0 < g_0 < g^*$, and that $w_0$ is a random vector with $\|w_0\| = 1$. Set $\eta_t = 1/g_t^2$ at all iterations. For any $0 < \varepsilon < 0.5$, let the learning rate of $g$ in Phase II satisfies*

$$0 < \gamma < \frac{g^* - g_0}{(1 - \varepsilon^2)(g^* - g_0) + \varepsilon^2 g^*} < 1. \tag{33}$$

*Let the number of iterations be*

$$T_1 = \frac{\log(1/\varepsilon^2)}{\log(g^{*2}/g_0^2)}, \quad T_2 = \frac{\log(\frac{1 - (1 - \varepsilon^2)g_0/g^*}{\varepsilon^2})}{\log(\frac{1}{1 - (1 - \varepsilon^2)\gamma})}. \tag{34}$$

*Then after $T = T_1 + T_2$ iterations, the output of Algorithm 2 will satisfy*

$$\langle w_T, w^* \rangle \geq 1 - \varepsilon, \quad (1 - 2\varepsilon^2)g^* \leq g_T \leq g^*, \tag{35}$$

*which indicates that $g_T w_T$ is close to the minimum $\ell_2$-norm solution $g^* w^*$. We can also bound the final loss as $f(w_T, g_T) = \|Ag_T w_T - Ag^* w^*\|^2/2 \leq 3\varepsilon g^{*2}$.*

Simplification for $T_1$ and $T_2$ (here we assume $\gamma < \frac{1}{4}$ to get the order in Theorem 3.2):

$$T_1 = \frac{\log(1/\varepsilon^2)}{\log(g^{*2}/g_0^2)} = \frac{\log(1/\varepsilon^2)}{\log(1 + (g^{*2} - g_0^2)/g_0^2)} \overset{(a)}{\approx} \frac{\log(1/\varepsilon^2)}{(g^{*2} - g_0^2)/g_0^2} = \frac{g_0^2}{\delta_0}\log(1/\varepsilon^2), \tag{36}$$

$$T_2 = \frac{\log\left((g^* - g_0)/(g^*\varepsilon^2) + g_0/g^*\right)}{\log\left(1 + (1 - \varepsilon^2)\gamma/(1 - (1 - \varepsilon^2)\gamma)\right)} \overset{(b)}{\approx} \frac{\log\left((g^* - g_0)/(g^*\varepsilon^2)\right)}{(1 - \varepsilon^2)\gamma/(1 - (1 - \varepsilon^2)\gamma)} \overset{(c)}{\approx} \frac{1}{\gamma}\log\left(\frac{g^{*2} - g_0^2}{g^{*2}\varepsilon^2}\right). \tag{37}$$

For step $(a)$ and $(b)$, we apply $\log(1 + x) \geq x \log 2, x < 1$ for denominator. For step $(b)$, we take out the constant term $g_0/g^*$ in the numerator. For step $(c)$ inside the $log$ term, we multiply $g^* + g_0$ for both numerator and denominator as follows

$$\frac{g^* - g_0}{g^*\varepsilon^2} = \frac{(g^* - g_0)(g^* + g_0)}{g^*(g^* + g_0)\varepsilon^2} \leq \frac{(g^{*2} - g_0^2)}{g^{*2}\varepsilon^2}.$$

*Proof.* For any vector $w \in \mathbb{R}^d$, we use $w^{\|} \in \mathbb{R}^d$ to denote its projection onto the row space of $A$. We use $w^{\perp} \in \mathbb{R}^d$ to denote its component in the subspace that is orthogonal to the row space of $A$. Since $A$ has orthogonal rows, we can write $w = w^{\|} + w^{\perp}$, where

$$w^{\|} = A^{\top}Aw, \quad w^{\perp} = (I - A^{\top}A)w. \tag{38}$$

Since $w^*$ is the minimum $\ell_2$-norm solution, $w^{*\perp}$ must be zero, i.e., $(I - A^\top A)w^* = 0$ and $A^\top A w^* = w^*$.

We will show that the algorithm has two phases. We now look at each phase in more detail.

**Phase I.** For any $t = 0, ..., T_1 - 1$, only $w$ is updated.

$$
\begin{aligned}
v_t &\overset{(a)}{=} w_t - \eta_t g_t^2 A^\top A w_t + \eta_t g_t g^* A^\top A w^* \\
&\overset{(b)}{=} (I - A^\top A)w_t + \frac{g^*}{g_0} A^\top A w^* \\
&\overset{(c)}{=} w_t^\perp + \frac{g^*}{g_0} w^*,
\end{aligned}
\tag{39}
$$

where (a) follows from substituting the partial gradient, (b) is true because of the choice of our learning rates: $\eta_t = 1/g_t^2$ and $\gamma_t = 0$, and (c) follows from the fact that $A$ has orthonormal rows. Since $w_t^\perp$ is orthogonal to $w^*$ and $g_0 < g^*$, we have

$$
\|v_t\|^2 = \|w_t^\perp\|^2 + g^{*2}/g_0^2 \geq g^{*2}/g_0^2 > 1.
\tag{40}
$$

After normalization, we have $w_{t+1} = v_t/\|v_t\|$. As shown in (39), gradient update does not[7] change the component in the orthogonal subspace: $v_t^\perp = w_t^\perp$. Since $\|v_t\|^2 > 1$, the orthogonal component will shrink after the normalization step:

$$
\|w_{t+1}^\perp\|^2 = \frac{\|v_t^\perp\|^2}{\|v_t\|^2} = \frac{\|w_t^\perp\|^2}{\|w_t^\perp\|^2 + g^{*2}/g_0^2} \leq \frac{g_0^2}{g^{*2}}\|w_t^\perp\|^2.
\tag{41}
$$

Since $g_0 < g^*$, after $T_1 = \frac{\log(1/\varepsilon^2)}{\log(g^{*2}/g_0^2)}$ iterations, we have

$$
\|w_{T_1}^\perp\|^2 \leq (g_0^2/g^{*2})^{T_1} \leq \varepsilon^2.
\tag{42}
$$

As indicated in (39), $w_t^\|$ is in the same direction as $w^*$ for $t \geq 1$. Since $\|w_{T_1}^\perp\| \leq \varepsilon$, $\|w_{T_1}^\|\| \geq \sqrt{1 - \varepsilon^2} \geq 1 - \varepsilon$. Therefore, $\langle w_{T_1}, w^* \rangle = \|w_{T_1}^\|\| \geq 1 - \varepsilon$.

**Phase II.** For iteration $t = T_1, ..., T_1 + T_2 - 1$, the algorithm updates both $w$ and $g$. The learning rate of updating $g$ is set as a constant $0 < \gamma < 1$. The gradient update on $g$ is

$$
\begin{aligned}
g_{t+1} &= g_t - \gamma g_t w_t^T A^\top A w_t + \gamma g^* w_t^T A^\top A w^* \\
&\overset{(a)}{=} g_t - \gamma g_t \|w_t^\|\|^2 + \gamma g^* \left\langle w_t^\|, w^* \right\rangle, \\
&\overset{(b)}{=} g_t - \gamma g_t \|w_t^\|\|^2 + \gamma g^* \|w_t^\|\|,
\end{aligned}
\tag{43}
$$

where (a) follows from the fact that $A$ has orthonormal rows and $w^*$ lies in the row space of $A$, and (b) is true because (39) implies that $w_t^\|$ is in the same direction as $w^*$ for $t \geq 1$.

We will now prove that the following two properties (see Lemma E.8) hold during Phase II:

- Property (i): $\|w_{t+1}^\perp\| \leq \|w_t^\perp\| \leq \varepsilon$.

- Property (ii): letting $\gamma' = \gamma(1 - \varepsilon^2)$, we have
$$
(1 - \gamma')g_t + \gamma' g^* \leq g_{t+1} \leq g^*.
$$

We will now finish the proof of Theorem E.7 using these two properties. After $T = T_1 + T_2$ iterations, by Property (i) and the same argument as in Phase I, we have $\langle w_T, w^* \rangle = \|w_T^\|\| \geq 1 - \varepsilon$. By Property (ii), we can rewrite the lower bound of $g_T$ as

$$
\begin{aligned}
g^* - g_T &\leq (1 - \gamma')(g^* - g_{T-1}) \\
&\leq (1 - \gamma')^{T_2}(g^* - g_{T_1}) \\
&\overset{(a)}{=} (1 - \gamma')^{T_2}(g^* - g_0) \\
&\overset{(b)}{\leq} 2\varepsilon^2 g^*,
\end{aligned}
\tag{44}
$$

where (a) follows from the fact that $g_{T_1} = g_0$, and (b) follows from our choice of $T_2$: it is easy to verify that $T_2$ satisfies $(1-\gamma')^{T_2}(g^* - g_0 + \delta) = \delta$, which implies that $(1-\gamma')^{T_2}(g^* - g_0) < \delta$. By our definition, $\delta = \varepsilon^2 g^*/(1-\varepsilon^2) < 2\varepsilon^2 g^*$ for $0 < \varepsilon < 0.5$. Therefore, by (44), we have $g_T \geq (1 - 2\varepsilon^2)g^*$.

Given $\langle w_T, w^* \rangle \geq 1 - \varepsilon$ and $(1 - 2\varepsilon^2)g^* \leq g_T \leq g^*$, we can bound the loss as

$$
\begin{aligned}
f(w_T, g_T) &= g_T^2 \|Aw_T\|^2/2 - g_T g^* \langle Aw_T, Aw^* \rangle + g^{*2}/2 \\
&\leq g^{*2}/2 - (1 - 2\varepsilon^2)g^{*2}(1 - \varepsilon) + g^{*2}/2 \\
&\leq 3\varepsilon g^{*2}.
\end{aligned}
$$

$\square$

### E.2.1 Technical Lemmas for Theorem E.7

**Lemma E.8.** *We have following property in Phase II for Theorem E.7*

- *Property (i): $\|w_{t+1}^\perp\| \leq \|w_t^\perp\| \leq \varepsilon$.*

- *Property (ii): letting $\gamma' = \gamma(1 - \varepsilon^2)$, we have*

$$
(1 - \gamma')g_t + \gamma' g^* \leq g_{t+1} \leq g^*.
$$

*Proof.* We will argue by induction. We first show that the above two properties hold when $t = T_1$. Since $g_{T_1} = g_0 < g^*$, by (41), we have $\|w_{T_1+1}^\perp\| \leq g_{T_1}\|w_{T_1}^\perp\|/g^* < \|w_{T_1}^\perp\| \leq \varepsilon$. By (43), we have

$$
\begin{aligned}
g_{T_1+1} &\overset{(a)}{\leq} g_{T_1} - \gamma g_{T_1}(1 - \varepsilon^2) + \gamma g^* \\
&= g_{T_1} - \gamma g_{T_1}(1 - \varepsilon^2) + \gamma(1 - \varepsilon^2)g^* + \gamma \varepsilon^2 g^* \\
&\overset{(b)}{=} g_0 - \gamma' g_0 + \gamma' g^* + \gamma' \delta \\
&= g^* - (1 - \gamma')(g^* - g_0) + \gamma' \delta, \tag{45}
\end{aligned}
$$

and

$$
\begin{aligned}
g_{T_1+1} &\overset{(c)}{\geq} g_{T_1} - \gamma g_{T_1}\|w_{T_1}^\|\|^2 + \gamma g^*\|w_{T_1}^\|\|^2 \\
&= g_{T_1} + \gamma(g^* - g_{T_1})\|w_{T_1}^\|\|^2 \\
&\overset{(d)}{\geq} g_{T_1} + \gamma(g^* - g_{T_1})(1 - \varepsilon^2) \\
&= (1 - \gamma')g_{T_1} + \gamma' g_{T_1}, \tag{46}
\end{aligned}
$$

where inequalities (a), (c), and (d) follow from the fact that $1 - \varepsilon^2 \leq \|w_{T_1}^\|\|^2 \leq 1$, and (b) follows from our definition $\gamma' = \gamma(1 - \varepsilon^2)$, $\delta = \varepsilon^2 g^*/(1 - \varepsilon^2)$, and the fact that $g_{T_1} = g_0 < g^*$. By the upper bound of $\gamma$ given in (33), we can verify that $(1 - \gamma')(g^* - g_0) > \gamma' \delta$, and hence, (45) implies that $g_{T_1+1} < g^*$.

Now suppose that Property (i) and (ii) hold for $t = T_1, ..., k - 1$, where $T_1 \leq k - 1 < T_1 + T_2 - 1$. We need to prove that they also hold for the $k$-th iteration. By assumption, $g_k \leq g^*$, so using the same argument as (40) and (41), we have $\|v_k\|^2 \geq 1$ and $\|w_{k+1}^\perp\| = \|w_k^\perp\|/\|v_k\| \leq \|w_k^\perp\| \leq \varepsilon$, where the last step follows from Property (i) at the $(k - 1)$-th iteration. Therefore, Property (i) holds for the $k$-th iteration.

To prove Property (ii), first note that by assumption, $1 - \varepsilon^2 \leq \|w_k^\|\|^2 \leq 1$. We can use the same argument as (46) to show that $g_{k+1} \geq (1 - \gamma')g_k + \gamma' g^*$. We can also use a similar argument as (45) to get

$$
g_{k+1} \leq g^* - (1 - \gamma')(g^* - g_k) + \gamma' \delta, \tag{47}
$$

where $\gamma' = \gamma(1 - \varepsilon^2)$ and $\delta = \varepsilon^2 g^*/(1 - \varepsilon^2)$. The above equation can be rewritten as

$$
\begin{aligned}
g^* - g_{k+1} + \delta &\geq (1 - \gamma')(g^* - g_k + \delta) \\
&\geq (1 - \gamma')^{k+1-T_1}(g^* - g_{T_1} + \delta) \\
&\overset{(a)}{\geq} (1 - \gamma')^{T_2}(g^* - g_0 + \delta) \\
&\overset{(b)}{=} \delta,
\end{aligned}
\tag{48}
$$

where (a) follows from the fact that $k \leq T_1 + T_2 - 1$, and (b) can be verified for our choice of $T_2$. Eq. (48) implies that $g_{k+1} \leq g^*$.

$\square$

## E.3  General A matrix

**Proposition E.9** (General Matrix $A$). *For a full rank matrix $A$ with $\lambda_{\max}(AA^\top) = 1$, we fix $\delta > 0$. In **Phase I** with fixed $g = g_0$ that satisfies $g_0 \leq \frac{g^*\lambda_{\min}(AA^\top)}{2+\delta}$ ,we can reach to a solution satisfying $\|w_{T_1}^\perp\| \leq \varepsilon$ where*

$$
T_1 = \frac{1}{\log(1 + \delta)} \log\left(\frac{\|w_0^\perp\|}{\varepsilon}\right).
$$

*Moreover, if the singular values of $A$ do not decrease too fast, so that the following inequality holds:*

$$
\frac{1}{m}\|AA^\top\|_F^2 \geq \lambda_{\min}^2(AA^\top) + 2\sqrt{\frac{\log(m)}{m}},
\tag{49}
$$

*and $w^*$ is randomly drawn on the sphere, then with probability $1 - \mathcal{O}(\frac{1}{m})$, we only need that*

$$
g_0 \leq \frac{g^*}{(2 + \delta)}\sqrt{\frac{\|AA^\top\|_F^2 - 2\sqrt{m\log(m)}}{m}}.
\tag{50}
$$

**Lemma E.10** (For all $w^*$). *Let $\sigma_i$ be the singular values of $A$ in decreasing order, let $r$ be the rank of $A$, so that $\sigma_r > 0$. We fix $g := g_0$ satisfying*

$$
g_0 \leq \frac{g^*\sigma_r}{2 + \delta - \sigma_r}
$$

*and update only $w$ using rPGD. Then we have the orthogonal component $w^\perp$ decreases geometrically such that $\|w_{T_1}^\perp\| \leq \epsilon$ after iteration*

$$
T_1 = \frac{1}{\log(1 + \delta)} \log\left(\frac{\|w_0^\perp\|}{\epsilon}\right)
$$

*Proof.* Consider the singular value decomposition of $A^\top A = U\Sigma U^\top$ with

$$
\Sigma = \begin{bmatrix} \sigma_1 & & & \\ & \ddots & & \\ & & \sigma_m & \\ & & & \mathbf{0}_{d-m} \end{bmatrix} \quad \text{with } 1 = \sigma_1 \geq \sigma_2 \geq \ldots \geq \sigma_m > 0.
\tag{51}
$$

Moreover $U$ is a $d \times d$ orthogonal matrix. We now use superscripts $t$ to illustrate the $t$th iteration $w_t$ since we use subscript for the eigenvalues index. Let $\eta = \frac{1}{g_t^2\sigma_1} = \frac{1}{g_t^2}$. The update of $v_t$ can be written as

$$
v_t = w_t - \eta g_0 A^\top A(g_0 w_t - g^* w^*) = (I - A^\top A)w_t + \frac{g^*}{g_0}A^\top Aw^* = U(I - \Sigma)U^\top w_t + \frac{g^*}{g_0}U\Sigma U^\top w^*
$$

$$\|v_t\| = \|\frac{g^*}{g_0}\Sigma U^\top w^* + (I - \Sigma) U^\top w_t\|$$

$$= \left\| \frac{g^*}{g_0}\begin{bmatrix} \sigma_1 & & \\ & \ddots & \\ & & \sigma_m \\ & & & \mathbf{0}_{d-m} \end{bmatrix} U^\top w^* + \begin{bmatrix} 0 & & \\ & \ddots & \\ & & 0 \\ & & & \mathbf{1}_{d-m} \end{bmatrix} U^\top w_t + \begin{bmatrix} 1 - \sigma_1 & & \\ & \ddots & \\ & & 1 - \sigma_m \\ & & & \mathbf{0}_{d-m} \end{bmatrix} U^\top w_t \right\|$$

$$\geq \sqrt{\left(\frac{g^*}{g_0}\right)^2 \sum_{i=1}^m \sigma_i^2 [U^\top w^*]_i^2 + \sum_{i=m+1}^d [U^\top w_t]_i^2} - \sqrt{\sum_{i=1}^m (1 - \sigma_i)^2 [U^\top w_t]_i^2}$$

$$\geq \frac{g^*}{g_0}\sigma_m - (1 - \sigma_m) \tag{52}$$

$$\geq \left(\frac{g^*}{g_0} + 1\right)\sigma_m - 1$$

$$\geq 1 + \delta$$

since we have

$$\sigma_m \geq \frac{2 + \delta}{\left(\frac{g^*}{g_0} + 1\right)} \quad \Leftrightarrow \quad g_0 \leq \frac{g^*\sigma_m}{2 + \delta - \sigma_m} \text{ and } \sigma_m \leq 2$$

Note that the singular values are sorted so that $\sigma_m \leq \sigma_1 = 1$, so the second inequality clearly holds. The above inequality implies that as long as we have $g_0$ small, we can always guarantee $\|v\| \geq 1 + \delta$. Using the equality

$$\|w_{t+1}^\perp\| = \frac{\|w_t^\perp\|}{\|v_{t+1}\|} \leq \frac{1}{1 + \delta}\|w_t^\perp\|$$

we see that the orthogonal component $w^\perp$ decreases geometrically. $\qquad \square$

**Lemma E.11** (random vector $w^*$ uniformly distributed on the sphere). *Suppose further that $w^*$ is randomly drawn on the sphere, i.e. $w^* = \frac{z}{\|z\|}$ where $z \sim \mathcal{N}(0, I_d)$. If the input data matrix $A$ satisfies :*

- *the maximum eigenvalue of $\lambda_{max}(AA^\top) = 1$*

- *the rank of $AA^\top$ is $m$.*

- *the spectral of $A \in \mathbb{R}^{m \times d}$ satisfies $\frac{1}{m}\left(\|AA^\top\|_F^2 - 2\sqrt{m\log(m)}\right) \geq \sigma_m^2$ where the $\sigma_m$ is the minimum eigenvalue of $AA^\top$.*

*Then we can fix $g := g_0$ satisfying*

$$g_0 \leq \frac{g^*}{(2 + \delta - \sigma_m)}\sqrt{\frac{\|AA^\top\|_F^2 - 2\sqrt{m\log(m)}}{m}}$$

*and update only $w$ using rPGD. Then with probability $1 - \mathcal{O}(\frac{1}{m})$, we have the orthogonal component $w^\perp$ decreases geometrically such that $\|w_{T_1}^\perp\| \leq \epsilon$ after iteration*

$$T_1 = \frac{1}{\log(1 + \delta)}\log\left(\frac{\|w_0^\perp\|}{\epsilon}\right)$$

*Proof.* Since $w^*$ is uniform on the $d$-dimensional sphere, $u^* := U^\top w^*$ (let $[U^\top w^*]_i = u_i^*$) is also uniform on the $d$-dimensional sphere. Moreover, we can represent the random vector $u^*$ as a standard Normal random vector divided by its norm, i.e.

$$u^* = \frac{z}{\|z\|},$$

where $z \sim \mathcal{N}(0, I_d)$. We want to lower bound $\sum_{i=1}^{m} \sigma_i^2 [U^\top w^*]_i^2 = \sum_{i=1}^{m} \sigma_i^2 (u_i^*)^2$. We can write

$$\sum_{i=1}^{m} (u_i^*)^2 \sigma_i^2 = \frac{\sum_i^m z_i^2 \sigma_i^2}{\sum_{i=1}^{m} z_i^2}.$$

Thus we need to get the upper bound of $\sum_{i=1}^{m} z_i^2$ and the lower bound of $\sum_{i=1}^{m} z_i^2 \sigma_i^2$. Note that

$$1 = \sigma_1 \geq \sigma_2 \geq \dots \sigma_r > 0.$$

Since $z_i \sim \mathcal{N}(0, 1)$, $X = \sum_{i=1}^{m} z_i^2$ is 1-sub-exponential r.v. with expectation 1. Thus, with Bernstein inequality (i.e., see Theorem 2.8.1 in Vershynin [2018]), for $\epsilon > 0$, we have that:

$$\mathbb{P}\left(\sum_{i=1}^{m} \sigma_i^2 z_i^2 \leq \sum_{i=1}^{m} \sigma_i^2 - \epsilon m\right) \leq \mathbb{P}\left(\left|\frac{1}{m}\sum_{i=1}^{m} \sigma_i^2 z_i^2 - \frac{1}{m}\sum_{i=1}^{m} \sigma_i^2\right| \leq \epsilon\right)$$

$$\leq 2 \exp\left(-c \min\left(\frac{\epsilon^2 m^2}{\sum_{i=1}^{m} \sigma_i^4}, \frac{\epsilon m}{\max_i \sigma_i^2}\right)\right)$$

$$\leq 2 \exp\left(-c \min\left(\frac{\epsilon^2 m^2}{\sum_{i=1}^{m} \sigma_i^2}, \frac{\epsilon m}{\max_i \sigma_i^2}\right)\right) \quad \text{since} \quad \sum_{i=1}^{m} \sigma_i^2 \geq \sum_{i=1}^{m} \sigma_i^4$$

$$(53)$$

and

$$\mathbb{P}\left(\sum_{i=1}^{m} z_i^2 \geq (1+\epsilon)m\right) \leq \mathbb{P}\left(\left|\frac{1}{m}\sum_{i=1}^{m}(z_i^2 - 1)\right| \geq \epsilon\right) \leq 2\exp(-c\min\{m\epsilon^2, m\epsilon\}). \quad (54)$$

where $c$ is an absolute constant.

Let $\epsilon = \sqrt{\frac{\log(m)}{m}} \leq 1$. Then, $\epsilon^2 m \leq \frac{\epsilon^2 m^2}{\sum_{i=1}^{m} \sigma_i^2} < \frac{\epsilon m}{\max_i \sigma_i^2} = \epsilon m$ since $2\sqrt{m \log m} < \sum_i \sigma_i^2$. Thus (53) and (54) can be simplified, respectively,

$$\mathbb{P}\left(\sum_{i=1}^{m} \sigma_i^2 z_i^2 \leq \sum_{i=1}^{m} \sigma_i^2 - \sqrt{m \log(m)}\right) \leq \exp\left(-cm\epsilon^2\right) = \frac{1}{m}e^{-c} = \mathcal{O}\left(\frac{1}{m}\right) \quad (55)$$

and

$$\mathbb{P}\left(\sum_{i=1}^{m} z_i^2 \geq m + \sqrt{m \log(m)}\right) \leq \frac{1}{m}e^{-c} = \mathcal{O}\left(\frac{1}{m}\right). \quad (56)$$

Then with probability $1 - \mathcal{O}\left(\frac{1}{m}\right)$, we have

$$\sum_{i=1}^{m} (u_i^*)^2 \sigma_i^2 = \frac{\sum_i^m z_i^2 \sigma_i^2}{\sum_{r=1}^{m} z_i^2} \geq \frac{\sum_{i=1}^{m} \sigma_i^2 - \sqrt{m \log(m)}}{m + \sqrt{m \log m}} \geq \frac{1}{m}\sum_{i=1}^{m} \sigma_i^2 - 2\sqrt{\frac{\log(m)}{m}} \geq \sigma_m^2$$

where the last inequality is due to the assumption that the spectral satisfies $\frac{1}{m}\sum_{i=1}^{m} \sigma_i^2 > 2\sqrt{\frac{\log(m)}{m}} + \sigma_m^2$. To sum up, with probability $1 - \mathcal{O}\left(\frac{1}{m}\right)$,

$$\sum_{i=1}^{m} \sigma_i^2 [u^*]_i^2 \geq \frac{1}{m}\sum_{i=1}^{m} \sigma_i^2 - 2\sqrt{\frac{\log(m)}{m}} = \frac{1}{m}\left(\|AA^\top\|_F^2 - 2\sqrt{m \log(m)}\right)$$

Now, using the derivation in (52) for lower bound of $\|v_t\|$, we have that:

$$\|v_t\| \geq \frac{g^*}{g_0}\sqrt{\frac{\|AA^\top\|_F^2 - 2\sqrt{m \log(m)}}{m}} - (1 - \sigma_m) \geq 1 + \delta$$

$$\Rightarrow g_0 \leq \frac{g^*}{(2 + \delta - \sigma_m)}\sqrt{\frac{\|AA^\top\|_F^2 - 2\sqrt{m \log(m)}}{m}}$$

With $g_0$ satisfying above, we can guarantee that $\|v_t\| \geq 1 + \delta$.

$\square$

## F Experiments

We evaluate WN and rPGD on two problems: linear regression and matrix sensing. Due to space limit, we only include the experiment for linear regression here and put the experiment for matrix sensing to the appendix and matrix sensing. We show that for a wide range of initialization, WN and rPGD converges to the minimum $\ell_2$-norm solution for linear regression and the minimum nuclear norm solution for matrix sensing. This is in contrast to the standard GD algorithm. For both problems GD requires initialization very close to, or exactly at, the origin to converge to the minimum norm solution Li et al. [2018]. We will compare with the following two step-size schemes.

(1) **Algorithm with $\gamma_t = \eta_t$**: We simultaneously update the weight vector (matrix) and the scalar $g$. This is similar to the training of deep neural networks, where we use the same learning rate for all of the layers.

(2) **Two-phase algorithm**: In **Phase I**, we use sufficiently small learning rate to update $g$, the scale component (a scalar in linear regression). In **Phase II**, we use large step-size to update $g$. For both phases, we use large learning rate to update the direction component (weight vector in linear regression and weight matrix in matrix sensing)

### F.1 Linear Regression

Let $m = 20$, $d = 50$. We generate the feature matrix as $A = U\Sigma V^T \in \mathbb{R}^{m \times d}$, where $U \in \mathbb{R}^{m \times m}$ and $V \in \mathbb{R}^{d \times m}$ are two random orthogonal matrices chosen uniformly over the Stiefel manifold of partial orthogonal matrices, and $\Sigma$ is a diagonal matrix described below. Let $\kappa = \frac{\lambda_{\max}(AA^\top)}{\lambda_{\min}(AA^\top)}$. We vary the condition number $\kappa \geq 1$ of $A$ in our experiments. The diagonal entries of $\Sigma$ are set as $1, (1/\kappa)^{1/(m-1)}, (1/\kappa)^{2/(m-1)}, ..., 1/\kappa$. Set $g^* = 3$, and $w^*$ as an arbitrary unit norm vector.

Let $w_0$ be a random unit norm vector. We run the standard gradient descent (GD) algorithm on the problem 1 with the initialization $x_0 = g_0 w_0$. We run Algorithm 1 and 2 starting from the same initialization, and plot $|\widehat{g}| = \|\widehat{g}\widehat{w}\|_2$ as a function of $g_0$. We run all of the algorithms until the squared loss satisfies $f(\widehat{w}, \widehat{g}) \leq 10^{-5}$, where the final solution is denoted as $\widehat{g}\widehat{w}$. We have the following observations:

**Figure 1** shows the result when we set a very small but equal learning rate for $w$ and $g$: $\eta_t = \gamma_t = 0.005$. It shows there is no difference between WN and rPGD when the learning rate is small, which matches Lemma 2.2. We can see that both WN and rPGD can get close to the minimum norm solution with a large range of initializations ($g^*w^*$ for $g_0 \lesssim 1.5$) while this is only true for GD when $g_0$ is close to 0. This experiment supports our theory.

**The top plot in Figure 3** shows the result when we set relatively large learning rates of $w$ and $g$: $\eta_t = \gamma_t = 0.1$, as in practice where we use the same non-vanishing learning rate for all the layers when training deep neural networks. The plot shows a difference between WN and rPGD when $g_0 > 2$, while the two perform similarly when $g_0 < 2$.

**The bottom plot in Figure 3** is when we set (1) WN $\eta_t = \|w_t\|/(g_t^2 \lambda_{\max})$ for $w$ and $\gamma_t = 0.005$ for $g$; (2) rPGD $\eta_t = 1/(g_t^2 \lambda_{\max})$ and $\gamma_t = 0.005$. This mimics the two-phase algorithm as shown in Theorem 3.2. We can arrive at a solution close to the minimum norm solution for even wider range of $g_0 \lesssim 3$.

**Robustness to the condition number $\kappa$.** We repeat the previous experiment for various input matrix $A$ with a wider range of $\kappa$ with fixed initialization $g_0 = 2.8$. The top plot in Figure 4 shows that for $\gamma_t = \eta_t$ as $\kappa$ increases, the $\ell_2$-norm of the solutions provided by WN and rPGD also gradually increases but not as much as those provided by the vanilla GD. The bottom plot in Figure 4 shows that the performance of the two-phase algorithms, with $\eta_t = 0$ in the first 5000 iterations, thus have a better performance compared with algorithm using $\gamma_t = \eta_t$.

### F.2 Experiment: Matrix Sensing

We show that the normalization methods can also be applied to the matrix sensing problem, to get closer to the minimum nuclear norm solutions. The goal in the matrix sensing problem is to recover a low-rank matrix from a small number of random linear measurements. Here we follow the setup considered in [Li et al., 2018] (for more related work on matrix sensing and completion, see,

Figure 3: **Fixed Orthogonal Matrix** $A$. Comparison of the final solutions $|\hat{g}| = \|\hat{g}\hat{w}\| = \|\hat{x}\|$ provided by GD, WN and our proposed rPGD on an overparameterized linear regression problem $\min_x \frac{1}{2}\|Ax - y\|_2^2$. All algorithms start from the same initialization $x_0 = g_0 w_0$. Compared to GD, WN and rPGD converge to the minimum $\ell_2$-norm solution for a larger region of initialization. Top plot is when we use the same stepsize for $w$ and $g$: $\gamma_t = \eta_t = 0.1$. Bottom plot is when we use a particularly small stepsize for $g$ and optimal stepsize for $w$. This implies that a small stepsize for $g$ can arrive to a solution that is close to the minimum-norm solution for even wider range of $g_0$.

e.g., Candès and Recht [2009], Donoho et al. [2013], Ge et al. [2016] and references therein). Let $X^* = U^* U^{*T} \in \mathbb{R}^{d \times d}$ (with $U^* \in \mathbb{R}^{d \times r}$) be the ground-truth rank-$r$ matrix. Let $A_1, .., A_m \in \mathbb{R}^{d \times d}$ be $m$ random sensing matrices, with each entry sampled from a standard Gaussian distribution. We are interested in the setting when $r \ll d$ and $m \ll d^2$. Given $m$ linear measurements of the form $\langle A_i, X^* \rangle$, let $U \in \mathbb{R}^{d \times d}$ be the variable matrix, we define the (over-parameterized) loss function as

$$f(U) = \frac{1}{2m} \sum_{i=1}^{m} (\langle A_i, UU^T \rangle - \langle A_i, X^* \rangle)^2. \tag{57}$$

It is proved in Li et al. [2018] that if $m = \tilde{O}(dr^2)$, then gradient descent on $f(U)$, when initialized very close to the origin, can recover the low-rank matrix $X^*$.

**WN.** To apply WN, we need to reparametrize $U$ into a direction variable and a scale variable. We consider two choices:

- Let $UU^T = g \frac{WW^T}{\|W\|_F^2}$, where $g \in \mathbb{R}$, and $W \in \mathbb{R}^{d \times d}$. In Figure 5, the green curve represents this algorithm. We label it with **WN**.

- Let $UU^T = WDW^T$, where $D \in \mathbb{R}^{d \times d}$ is a diagonal matrix, and all the column vectors of $W \in \mathbb{R}^{d \times d}$ have unit $\ell_2$ norm. That is, for $w_i \in \mathbb{R}^d, i = 1, 2, \ldots, d$

$$W = \left[ \frac{w_1}{\|w_1\|}; \frac{w_2}{\|w_2\|}; \ldots; \frac{w_d}{\|w_d\|} \right].$$

Figure 4: **Various General Matrix** $A$. Fix $g_0 = 2.8$ and increase the condition number $\kappa$. Top plot: $\gamma_t = \eta_t = 0.01$. The $\ell_2$-norm of the WN and rPGD solutions increases slowly as $\kappa$ increases, but their norm is smaller than when using Gradient Descent. Bottom plot: $\gamma_t = 1, \eta_t = 0.1 \times \mathbb{1}_{\{t>5000\}}$. The $\ell_2$-norm of WN and rPGD solutions are robust to condition number and close to the minimum $\ell_2$-norm for any $\kappa$. Note that green, orange and black curves of the bottom plot overlap.

In Figure 5, the purple curve represents the algorithm. We label it with **WN-Diag** where "Diag" references the diagonal matrix.

**rPGD.** To apply rPGD , we need to reparametrize $U$ into a direction variable and a scale variable. We consider two choices:

- Let $UU^T = gWW^T$, where $g \in \mathbb{R}$, and $W \in \mathbb{R}^{d \times d}$ satisfies $\|W\|_F = 1$. See Algorithm 3. In Figure 5, the orange curve represents the algorithm. We label it with **rPGD**.
- Let $UU^T = WDW^T$, where $D \in \mathbb{R}^{d \times d}$ is a diagonal matrix, and all the column vectors of $W \in \mathbb{R}^{d \times d}$ are projected to have unit $\ell_2$ norm. See Algorithm 4. In Figure 5, the red curve represents the algorithm. We label it with **rPGD-Diag**.

---

**Algorithm 3 rPGD** for matrix sensing loss $f(W, g)$

---

**Input:** initialization $W_0$ and $g_0$, number of iterations $T$, step-sizes $\gamma_t$ and $\eta_t$.
**for** $t = 0, 1, 2, \cdots, T - 1$ **do**
    $V_t = W_t - \eta_t \nabla_W f(W_t, g_t)$
    $W_{t+1} = \frac{V_t}{\|V_t\|_F}$
    $g_{t+1} = g_t - \gamma_t \nabla_g f(W_t, g_t)$
**end for**

---

Denote the corresponding loss functions for rPGD as $f(W, g)$ and $f(W, D)$. Let $Z_0 = Z/\|Z\|_F$ where $Z$ is a matrix with i.i.d. Gaussian entries, after which all column vectors have been normalized. We set the experiments with the following initialization:

**Algorithm 4 rPGD-Diag** for matrix sensing loss $f(W, D)$

---

**Input:** initialization $W_0$ and $D_0$, number of iterations $T$, step-sizes $\gamma_t$ and $\eta_t$.
**for** $t = 0, 1, 2, \cdots, T - 1$ **do**
    $V_t = W_t - \eta_t \nabla_W f(W_t, D_t)$
    $W_{t+1} = V_t$ with all column vectors normalized.
    $\text{diag}(D_{t+1}) = \text{diag}(D_t) - \gamma_t \, \text{diag}(\nabla_D f(W_t, D_t))$
**end for**

---

Figure 5: Comparison of the final solutions $\|\widehat{X}\|_*$ provided by GD, WN, and rPGD on an overparameterized matrix sensing problem ((57)). All algorithms start from the same initialization with the scale $\alpha = \sqrt{\|UU^\top\|_F}$. Compared to GD, WN and rPGD converge close to the minimum nuclear-norm solution for a broader region of initialization. The left plot is when we use the same stepsize for $W$ and $g$: $\gamma_t = \eta_t = c$. The right plot is when we use $\eta_t = c$ and $\gamma_t = c \mathbb{1}_{\{t>1000\}}$. This suggests that Two-phase algorithm can arrive to a solution closer to the nuclear-norm solution for a broader range of $g_0$. The blue, green, and black curves of the top plot overlap when $0 < \alpha < 0.1$. The blue, orange, green, purple, and black curves of the bottom plot overlap when $0 < \alpha < 0.1$.

- For vanilla GD on $f(U)$, let $U_0 = \alpha Z_0$;
- For WN and rPGD, let $W_0 = Z_0$, and $g_0 = \alpha^2$ ;
- For WN-Diag and rPGD-Diag, let $W_0 = Z_0$ and $D_0 = \alpha^2 I$.

We set $d = 30$, $r = 4$, and $m = 60$. We simulate $y_i = \langle A_i, \hat{U}\hat{U} \rangle$ with $\hat{U} \in \mathbb{R}^{d \times r}$ generated as a random matrix.[8]

We compare the performance of gradient descent, and our algorithms for several initializations scales $g$. We run each algorithm until convergence (i.e., when the squared loss is less than $10^{-6}$).

Similar to Figure 3, we use different learning rate schemes to get the final solution. We use grid search to find appropriate constant learning rate c.[9] The top plot in Figure 5 uses the following learning rate: constant $c$ for gradient descent; $\eta_t = \gamma_t = c$ for rPGD (Algorithm 3 and 4); and set $\eta_t = \gamma_t = c\|W\|_F$ for WN. The bottom plot in Figure 5 uses the two phase learning rates: constant for gradient descent; $\eta_t = c$ and $\gamma_t = c\mathbb{1}_{\{t>1000\}}$ for rPGD (Algorithm 3 and 4); and set $\eta_t = \gamma_t = c\|W\|_F$ and $\gamma_t = c\mathbb{1}_{\{t>1000\}}$ for WN. Compared to GD, WN and rPGD converge close to the minimum nuclear-norm solution for a larger region of initialization. Moreover, these results also suggest that with the two-phase algorithm, one can arrive to a solution close to the nuclear-norm solution for a wider range of $g_0$.

## Footnotes

[4]Note this is not an assumption. $\|Ag^* w^*\|^2 - \|A(g_0 w_0 - g^* w^*)\|^2 > 0$ is true from the our assumed condition $\langle Aw_0, Aw^* \rangle > 0$. See the lemma's proof for details.

[5] we could prove similarly for the case $g_0 < 0$ and $\langle Aw_0, Ag^*w^*\rangle < 0$

[6]Note that one could use $\frac{1}{2}\delta_0$ for the convenience of the proof.

[7]This is always true for linear regression because the gradient $\nabla_w f(w, g)$ lies in the row space of $A$.

[8]Code: $\hat{U} =$ numpy.random.randn$(d, r)$. Note that this is not necessary the minimum nuclear solution. We use the python package "cvxpy" to solve for the minimum nuclear solution for (57).

[9]Note $c$ varies for different $g_0$ and different algorithms. Here we start with $0.5$ and then decay by a factor of 2 until we get a step-size that can converge to the solution.