[Reviews · NeurIPS 2020]

Review 1

Summary and Contributions: This paper studied overparametrized linear regression problem and proposed a corresponding optimization algorithm with weight normalization. This is done by splitting the norm and the unit vector of weight vector and performing a projection during gradient descent.

Strengths: The authors studied the dynamics of proposed method for this specific problem in both continuous and discrete case.

Weaknesses: 1. The linear regression problem has a theoretical solution. It is pointless to study the gradient descent method in this setting. Also, the proposed method, on the contrary to standard weight normalization [52], can not generalize to nonlinear or higher dimension case. >> addressed in rebuttal 2. The experiments show that the proposed method derives a similar/smaller final norm compared to standard weight normalization. However, this does not prove that the method is useful. In fact, it only shows that this method provides a stronger constraint on the norm of the weight. >> addressed in the rebuttal

Correctness: Yes

Clarity: Yes, the method is clearly demonstrated.

Relation to Prior Work: It cited several related papers.

Reproducibility: Yes

Additional Feedback:


Review 2

Summary and Contributions: The paper considers implicit regularization through weight normalization and one similar heuristic called reparameterized projection gradient descent in underdetermined linear least square minimization. First, a continuous time analysis proves that the two heuristics are essentially equivalent since they have the same limit behavior when the stepsize parameters involved tend to zero. Then the asymptotic behavior of the limit flow is fully described as the stationary points are characterized and two regimes in the rate of convergence are identified depending on the norm of the initialization of the scalar parameter. Then, the discrete time analysis of the convergence properties is applied to the second heuristic (RGPD) first for orthogonal systems before being extended to nonorthogonal. A thorough discussion is provided to discuss limitations and possible extensions to nonlinear LS.

Strengths: Understanding implicit regularization is probably the most important topic to develop further the theoretical foundations of high dimensional inference and machine learning. I found that the paper offers significant results on one of the currently used heuristics, namely weight normalization. The authors have also reached rate of convergence results with nontrivial conditions and characterization. I see it as a strength of the paper to have considered such a simple framework to highlight such a sophisticated analysis. It's definitely one of the best papers I have reviewed or meta-reviewed for NeurIPS in the last three years.

Weaknesses: No weakness found.

Correctness: Arguments, method and statements are perfectly sound and well founded.

Clarity: The paper is very clear and as much as possible self-contained. Proofs of theorems are deferred to the Appendix. I appreciated a lot the fact that the authors focused on a simple use case (namely LS with linear models) to highlight highly nontrivial behavior in weight normalization heuristics. In general, I enjoyed the paper very much!

Relation to Prior Work: The positioning of the paper is well grounded on previous work. The list of references is quite impressive and accurately used along the paper. Great work!

Reproducibility: No

Additional Feedback:


Review 3

Summary and Contributions: This work studies the implicit bias of weight normalization for GD learning interpolating least square. In this setting, this work shows that gradient flow and GD with a dedicated learning rate schedule can converge to minimal normal solution, which is different from vanilla gradient flow / GD, where they find closest solution to initialization.

Strengths: This work extends our understanding for the implicit bias of weight normalization.

Weaknesses: Overall I think this is a really good submission. If I have to mention something, the analysis is limited to linear models, and cannot be applied to SGD -- but by no means we shall ask for more as an initial work to explain the implicit bias of weight normalization.

Correctness: Looks sound to me.

Clarity: Yes, the paper is very well organized and the different between WN/rPGD and GD are clearly explained. They included a very specified discussion with related works. I have enjoyed reading the paper.

Relation to Prior Work: Yes.

Reproducibility: Yes

Additional Feedback: For vanilla GD/gradient flow solving least square, the well know implicit bias says the algorithms would find the closest solution to initialization, since GD never updates the projection of the parameter to the orthogonal complement of the data matrix. However for WN/rPGD, this work (lemma 2.5) shows that the projection can move and even decrease with further assumptions. Thus WN/rPGD finds minimal norm solution. The similar normalized gradient methods have been studied before but only in terms of convergence -- this work is the first to address the regularization effects. I especially like the ODE analysis, which provides much insight why WN/rPGD can move the projection. Overall I think this is a very good work and would definitely suggest an acceptance. Typos: - Line 119: you mean "non-increasing"? - Line 187: it should be "w[2] = +1, -1". Small questions: - Stochastic version: to my understanding, your analysis heavily depends on computations related to the data matrix A^T A. For SGD with WN, however, A_i ^T A_i might not commute thus cannot be diagonalized simultaneously. Due to this reason I guess it is quite non-trivial to extend your analysis to SGD with WN. Please correct me if not. If so, can you comment on the implicit bias of SGD with WN, which is a more practical optimizer? - The implicit bias for GD actually holds for quite a general class of losses in additional to l2-loss. Can you comment on your results for other losses, e.g., l4-loss and exponential loss? ----- Post rebuttal ----- I have no questions and think the paper deserves an acceptance. I think the other reviewers should appreciate more about the theoretical contribution of this work.


Review 4

Summary and Contributions: This work presents a study on the implicit regularization and convergence for weight normalization. There have been numerous prior work talking on the implicit regularization of BN and also SGD itself, so the impact of this study is only marginal. As for the theoretically derivation, its loss function is over-simplified, and actually does not apply generally as the authors stated. --------------Post rebuttal comments--------------- The authors studied the regularization, convergence rate and stationary point of WN under the linear low-rank optimization setting.

Strengths: The theoretically analysis on the normalization methods, either its implicit regularization effect or its learning dynamics is always of importance for the community. This work presents WN as reparametrized projected gradient descent and also conducts both continuous and discrete-time analysis on its learning dynamics. Especially the latter is of interest. However, its general contribution is limited and its mathematical derivation would quickly fail to explain as we approach realistic learning situations.

Weaknesses: 1. The loss function chosen by the authors for the theoretical analysis is over-simplified and does not generally apply. For example, this work even does not take weight decay into consideration. Actually the statement that WN is equivalent to rPGD does not hold with weight decay since WN would suffer instability at w=0 while rPGD does not. 2. The implicit regularization effect of weight projection has already been talked previously in e.g. arxiv:1710.02338. The conclusion from this study is only marginal. 3. In the appendix A, the implicit regularization terms for both WN and rPGD are given. But the discussion on whether the term is larger than 0 is missing! Missing such discussion is a strong flaw to the main conclusion. 4. There are no empirical support for the conclusions.

Correctness: As stated above, there are some flaws in terms of its derivation.

Clarity: The paper is poorly written and is hard to follow its central idea.

Relation to Prior Work: Some related work are not mentioned in the paper, e.g. arxiv:1710.02338.

Reproducibility: Yes

Additional Feedback:

[Author Response · NeurIPS 2020]

**Reviewer 1 and 4**

We thank for the reviews and will resolve the main concerns. We sincerely ask the reviewers to re-evaluate the rating.
- Reviewer 1: "It is pointless to study the gradient descent method in this setting. Also, the proposed method, on the contrary to standard weight normalization [52], can not generalize to nonlinear or higher dimension case."
- Reviewer 4:"The loss function for the theoretical analysis is over-simplified, even does not take weight decay into consideration. The statement that WN is equivalent to rPGD does not hold with weight decay since WN would suffer instability at w=0 while rPGD does not.

Linear regression is a fundamental theoretical problem. When applying WN to linear regression, it becomes *non-convex* optimization. Moreover, the most important part in our setting is "under-determined".

We apologize for our word "proposed" in the paper and will remove this word. We would like to emphasize that our paper is **NOT** about proposing a new method (i.e., rPGD) but to **theoretically** understand the implicit regularization effect of these methods. rPGD is an existing method [13]. We build a surprising connection between rPGD and WN, which is exact equivalence under some condition (see Lemma 2.2). With infinitely small step-size $\eta \to 0$ and initialization $\|w_0\| = 1$, the equivalence of gradient flows of the two methods under the nonlinear or high dimension case will be maintained. However, when the stepsize is not small, the two methods are not the same as the norm $\|w_t\|$ grows for WN, while $\|w_t\| = 1, \forall t > 0$ for rPGD. Since we focus on implicit regularization, we do not want to involve the growing norm $\|w_t\|$ and so study rPGD, not WN.

We did not consider "weight decay" as our motivation is to study implicit regularization (IR) along the lines of [22]. Understanding algorithms **without** explicit regularization is the starting point for studying IR. If weight decay is used for linear regression, the problem becomes strongly convex and has a unique solution. However, in future work the referee's suggestion may be interesting as WN makes this setting (linear regression with weight-decay) non-convex.
[13] Douglas, Amari, Kung. "On gradient adaptation with unit-norm constraints." IEEE TSP 48.6 (1998): 1843-1847.
[22] Gunasekar, Suriya, et al. "Implicit regularization in matrix factorization." NeurIPS 2017.

- Reviewer 1: "The experiments show that the proposed method derives a similar/smaller final norm compared to standard weight normalization. However, this does not prove that the method is useful. In fact, it only shows that this method provides a stronger constraint on the norm of the weight."
- Reviewer 4: "There are no empirical support for the conclusions."

We would like to explain that our experiments are to support the theory and not to show the "usefulness" of rPGD. We want to show the implicit regularization along the research line [22]. You are right that the rPGD is likely to provide a stronger constraint on the norm of the weight.

- Reviewer 4: "The implicit regularization effect of weight projection has been talked previously in e.g. arxiv:1710.02338. This study is only marginal. In the appendix A, the discussion on whether the term is larger than 0 is missing!"

Thanks for the reference. We do not agree that our study is only a marginal improvement. Thanks for pointing out the discussion on whether the term is larger than 0. We now address here and will add to the paper. The regularization parameters are highly dependent on $g_t$, $g_{t+1}$ and the input matrix $A$. However, it is difficult to characterize the behavior of $\lambda_t$ in general. In particular, we require the parameters $g_t$, $g_{t+1}$, $w_t$ and $w_{t+1}$ updated in a way that $\lambda_t > 0$. For the simpler setting of orthogonal $A$, we can see for rPGD that: 1) If the learning rate of $g$ is small enough, we will have $g_{t+1} < g_t\|v_t\|$, which means that $\lambda_t > 0$; 2) When $g_t w_t$ is close to $g^* w^*$, we will have $\|v_t\| \approx 1$, and $g_{t+1} \approx g_t$, which means that $\lambda_t \approx 0$.

**Reviewer 2 and 3:**

We thank the reviewers for the positive evaluation.
- "Your analysis heavily depends on the data matrix $A^T A$. For SGD with WN, however, $A_i^T A_i$ might not commute thus cannot be diagonalized simultaneously. Due to this reason I guess it is quite non-trivial to extend your analysis to SGD with WN. Can you comment on the implicit bias of SGD with WN, which is a more practical optimizer?"

Indeed, it's not trivial to extend the continuous-time analysis to SGD with WN as we need to look at $w^\perp$, which depends on $A$. It is very challenging to analyse the discrete-time SGD case when updating both $g$ and $w$, because $g$ and $w$ are random variable and, by taking expectation, their product is hard to analyze. A possible alternative may be to look at the stochastic Langevin dynamics or making $g$ fixed.

- "The implicit bias for GD actually holds for quite a general class of losses in additional to l2-loss. Can you comment on your results for other losses, e.g., l4-loss and exponential loss?"

This is great question, we have thought about this. For $L_p$ loss, we need to think about the what norm should be used for the weight norm algorithm. With $L_4$ norm for the WN algorithm and $L_4$ loss, then $w_t^{\circ(3)}$ ($\circ$ is Hadamard power) is involved and the norm $\|w_t\|_4$ is no longer constant, which makes the dynamics harder to analyse.

[Meta-Review · NeurIPS 2020]

The paper was reviewed by experts on the topic and discussed after authors rebuttal. Results were found to be interesting and valuable. The reviewers comments should be taken into account while preparing the final version of the paper.